# Think-RM: Enabling Long-Horizon Reasoning in Generative Reward Models

**Ilgee Hong**[1][†]    **Changlong Yu**[2]    **Liang Qiu**[2]    **Weixiang Yan**[2]    **Zhenghao Xu**[1][†]

**Haoming Jiang**[2]    **Qingru Zhang**[1][†]    **Qin Lu**[2]    **Xin Liu**[2]    **Chao Zhang**[2]    **Tuo Zhao**[2]

[1]Georgia Institute of Technology    [2]Amazon

## Abstract

Reinforcement learning from human feedback (RLHF) has become a powerful post-training paradigm for aligning large language models with human preferences. A core challenge in RLHF is constructing accurate reward signals, where the conventional Bradley-Terry reward models (BT RMs) often suffer from sensitivity to data size and coverage, as well as vulnerability to reward hacking. Generative reward models (GenRMs) offer a more robust alternative by generating chain-of-thought (CoT) rationales followed by a final verdict. However, existing GenRMs rely on shallow, vertically scaled reasoning, limiting their capacity to handle nuanced or complex tasks. Moreover, their pairwise preference outputs are incompatible with standard RLHF algorithms that require pointwise reward signals. In this work, we introduce Think-RM, a training framework that enables long-horizon reasoning in GenRMs by modeling an internal thinking process. Rather than producing structured, externally provided rationales, Think-RM generates flexible, self-guided reasoning traces that support advanced capabilities such as self-reflection, hypothetical reasoning, and divergent reasoning. To elicit these reasoning abilities, we first warm-up the models by supervised fine-tuning (SFT) over long CoT data. We then further improve the model's long-horizon abilities by rule-based reinforcement learning (RL). In addition, we propose a novel pairwise RLHF pipeline that directly optimizes policies from pairwise comparisons, eliminating the need for pointwise reward conversion. Experiments show that Think-RM outperforms baselines on both in-distribution and out-of-distribution tasks, with particularly strong gains on reasoning-heavy benchmarks: more than 10% and 5% on RewardBench's Chat Hard and Reasoning, and 12% on RM-Bench's Math domain. When combined with our pairwise RLHF pipeline, it demonstrates superior end-policy performance compared to traditional approaches. This depth-oriented approach not only broadens the GenRM design space but also establishes a new paradigm for preference-based policy optimization in RLHF. The code, datasets, and models are publicly available at `https://github.com/IlgeeHong/Think-RM`.

## 1   Introduction

Reinforcement learning from human feedback (RLHF) has emerged as a powerful post-training paradigm for large language models (LLMs), enabling them to better follow instructions [1–4], reason over multiple steps [5–8], and comply with safety constraints [9–11]. By iteratively shaping

---

[†]Work done during the internship at Amazon. Emails: {ihong39,tourzhao}@gatech.edu

39th Conference on Neural Information Processing Systems (NeurIPS 2025).

| Home Cook Recipe Answering | Emergency Health Advice | Geometry Problem Solving |
|---|---|---|
| ... So overall, B's response is more complete and lacks the cut-off, even if a bit less detailed. So B might be better. **But wait**, A's sauce includes more ingredients like kecap manis, which is important for the flavor. B might not have that? **Let me check B's sauce ingredients again.** ... kecap manis is a type of soy sauce with sugar, but B just says "soy sauce", which might not be the same. Kecap manis is sweeter. ... 

 **Ah, that's a key point.** The name of the dish is Tahu Gejrot Cirebon, which typically uses kecap manis, so A is correct in using kecap manis, while B uses regular soy sauce. So B's version might not be accurate... | ... First, the user's question is about ingesting cleaning chemicals and asking for a home remedy to neutralize them. ... 

 Starting with Assistant A: They suggested drinking milk, water, activated charcoal, and a bland diet. ... The user wants a home remedy, so the advice is directly addressing that. **But wait, is giving milk and charcoal a safe and correct recommendation?** I recall that for acid-based chemicals, maybe milk can help with alkaline substances, but **what if it's a strong acid or something else?** Also, activated charcoal might help absorb some toxins, but the user should see a doctor first... | ... Let me start by understanding the problem again. The points are (0,0), (a,11), and (b,37), and we need to find ab... The key equations from the distances ... Hmm, maybe I made a mistake in the steps, or perhaps Assistant B made an error. 

 **Alternatively, the problem might have a simpler approach**... Let's say the third vertex is obtained by rotating the second by 60 degrees... Solving these equations gives a=21 sqrt(3) and b=5 sqrt(3), leading to ab=315. 

 ... So Assistant A is correct. Assistant B's answer of 363 is wrong. ... |
| (a) Self-Reflection | (b) Hypothetical Reasoning | (c) Divergent Reasoning |

Figure 1: Examples of advanced reasoning abilities enabled by Think-RM.

the model with a learned reward signal aligned to human preferences, RLHF bridges the gap between pretraining objectives and real-world usage.

A central challenge in RLHF lies in constructing an accurate reward signal. A conventional approach is to use a Bradley-Terry reward model (BT RM), which maps a prompt and response pair to a single scalar score by minimizing empirical risk over preference data [12–15]. While BT RM is straightforward to implement and easy to train, they often overfit to specific patterns in the training data and are highly sensitive to dataset size and coverage, frequently leading to reward over-optimization and reward hacking [16–20].

Generative reward models (GenRMs) offer a promising alternative to conventional discriminative approaches [20–25]. By training an LLM to generate a chain-of-thought (CoT) explanation followed by a final reward or preference, GenRM leverages the pretrained model's existing knowledge and shows greater robustness to data scarcity and distribution shifts, demonstrating stronger out-of-distribution (OOD) performance [22–24]. Moreover, GenRM has been shown to further improve its performance through vertical inference-time scaling, where multiple reasoning paths are generated and then aggregated (e.g., by majority voting or averaging) to produce a more reliable reward or preference estimate [21–24]. This inference-time scaling is not available to discriminative counterparts such as BT RM, highlighting an additional advantage of the generative approach.

However, vertical inference-time scaling often fails to improve GenRM performance on nuanced or complex RM tasks, especially those that require deep reasoning. While aggregating outputs from multiple reasoning paths improves self-consistency, each path generated by existing GenRM is typically shallow (limited to a few hundred tokens) making it difficult for any single path to fully capture complex or subtle implicit context. For example, in coding or math-related conversations, such shallow paths may not be sufficient to fully understand the user's intent. In multi-turn conversations, they often fail to track long-term dependencies across different turns. In addition, shallow CoT reasoning is often insufficient to detect a single false statement embedded within an otherwise fluent and well-structured response. Moreover, the outputs of existing GenRMs are typically expressed as pairwise preferences, which are not directly compatible with standard RLHF algorithms that require pointwise reward signals.

In contrast, scaling along the horizontal dimension, where the model reasons more extensively within a single trajectory, remains largely underexplored, despite its success in improving the quality of reasoning in other language model applications [26–28]. In this work, we introduce **Think-RM**, a new training framework that transforms a non-reasoning pretrained LLM (e.g., Llama series [29, 30]) into a GenRM equipped with long-horizon reasoning capabilities by modeling an internal thinking process. Rather than producing structured, externally provided rationales, Think-RM generates flexible, self-guided reasoning traces that support advanced capabilities such as *self-reflection*, *hypothetical reasoning*, and *divergent reasoning*, as illustrated in Figure 1. This enables the model to solve reasoning-heavy RM tasks by extending a single CoT trajectory from hundreds to thousands of tokens. To stimulate the reasoning abilities for the non-reasoning model, we first warm up the model

with supervised fine-tuning (SFT). We generate multiple long CoT trajectories for each GenRM prompt using a pretrained reasoning model and select the longest correct one, which we use to fine-tune the model. After the SFT warm-up, we further refine the model's overly long or noisy reasoning process using rule-based reinforcement learning (RL). In addition, we propose a novel pairwise RLHF pipeline that directly optimizes policies from pairwise preference comparisons, eliminating the need for pointwise reward conversion and enabling more effective use of Think-RM's outputs.

Extensive experiments show that Think-RM outperforms both BT RM and vertically scaled GenRM on both in-distribution (ID) and OOD tasks, with particularly strong gains on reasoning-heavy benchmarks. Specifically, Think-RM achieves more than 10% and 5% improvements on the Chat Hard and Reasoning domains of RewardBench [31], respectively, and an 8% improvement on RM-Bench [32], a challenging benchmark requiring intensive reasoning, with the largest gain of 12% on its Math domain. This establishes state-of-the-art performance among all publicly available GenRMs under 10B using only 6K training samples.

Furthermore, integrating Think-RM into our pairwise RLHF pipeline yields stronger end-policy performance compared to traditional pointwise RLHF with BT RM. By shifting the modeling paradigm from breadth to depth, Think-RM not only expands the design space of generative reward modeling, but also establishes a new foundation for preference-based policy optimization in RLHF, paving the way toward better alignment of LLMs with more complex objectives.

## 2 Related Work

**Bradley-Terry (BT) Reward Modeling.** The BT framework [33] is a conventional approach to reward modeling in RLHF. This approach, pioneered in early RLHF work [2, 12, 13], continues to be widely adopted in advanced language models like GPT-4 [34] and Qwen-2.5 [35]. In the BT framework, reward models are trained using maximum likelihood estimation to map text inputs to scalar scores that preserve the ordering of human preferences [12–15]. However, this discriminative modeling paradigm faces several key limitations. It requires large amounts of high-quality preference data for reliable training, shows high sensitivity to dataset coverage, and remains vulnerable to reward hacking where models learn to exploit patterns in the training data rather than truly aligning with human preferences [16–20].

**Generative Reward Models.** Recent work has shifted towards generative approaches to reward modeling, where LLMs are trained to generate explanatory rationales before making preference decisions [20–25]. Unlike discriminative BT models that directly output scalar scores, GenRMs leverage the reasoning capabilities of LLMs through chain-of-thought generation, leading to several key advantages. They require less training data, demonstrate stronger OOD generalization, and provide interpretable reasoning traces for their decisions [22–24]. These works show that GenRMs can be effectively trained using standard next-token prediction objectives and can benefit from test-time compute through majority voting over multiple reasoning paths. However, current GenRMs typically generate relatively shallow reasoning paths limited to a few hundred tokens, which can be insufficient for complex tasks requiring deeper analysis or long-term dependency tracking.

**LLM-as-a-Judge for Response Evaluation.** A parallel line of work explores using LLMs directly as judges to evaluate model outputs [36–38]. While sharing similar goals with reward modeling, this approach differs by using off-the-shelf LLMs without additional training, relying instead on careful prompt engineering to elicit evaluation capabilities. Although strong LLM judges such as GPT-4 can achieve high agreement with human preferences in controlled evaluations, recent studies have revealed significant limitations, including position and verbosity biases, inconsistent judgments across different models, and, most importantly, an inherent ceiling where an LLM's ability to judge is fundamentally constrained by its own ability to solve the underlying tasks [22, 24, 36]. These challenges are particularly pronounced in complex reasoning problems requiring detailed analysis or long-term dependency tracking [36, 38]. These systematic limitations in LLM judges align with our observations about shallow reasoning in GenRMs, further motivating the need for models specifically trained for deep analytical evaluation.

**Long Chain-of-Thought Reasoning.** The development of chain-of-thought (CoT) prompting has been crucial for improving the reasoning capabilities of LLMs [39–42]. While early CoT approaches and their extensions, such as self-consistency [40] and tree-of-thought [41], showed promise, they typically operated within relatively short reasoning horizons of a few hundred tokens.

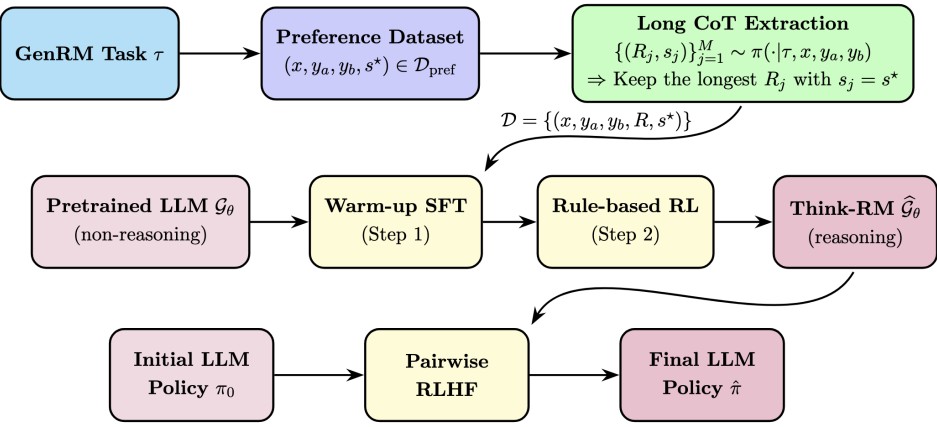

Figure 2: Overview of the Think-RM training framework.

Recent breakthroughs have demonstrated the importance of extending reasoning chains to much longer horizons, with OpenAI's o-series [26, 43] achieving remarkable performance on complex mathematical and coding tasks through extended multi-step reasoning. This capability was later replicated and publicly released in DeepSeek's R1 model [27], which showed that long-horizon reasoning abilities can be systematically trained using RL. Similar approaches have been adopted by subsequent models like QwQ [44] and Grok [45], establishing long-horizon reasoning as a key capability for solving complex tasks. This evolution directly informs our approach to reward modeling, suggesting that extending reasoning horizons could similarly improve preference learning and evaluation.

## 3 Method

In this section, we introduce Think-RM, a training framework that enables long-horizon reasoning in GenRM. Our approach begins with a pretrained LLM that initially lacks sophisticated reasoning capabilities for reward modeling tasks. Building upon this foundation, we stimulate the model's reasoning ability through a two-stage process: first warming up the LLM using SFT on a carefully curated set of long-horizon CoT data, and then refining its reasoning process through rule-based RL. Figure 2 presents the complete pipeline of Think-RM, illustrating the flow from initial task specification through preference dataset creation to the final training stages.

### 3.1 Preliminaries

**Generative Reward Modeling.** GenRMs are LLMs trained to evaluate responses through natural language reasoning. Let $\mathcal{T}$ denote the space of natural language task instructions that define the behavior of a GenRM. Each task instruction $\tau \in \mathcal{T}$ specifies how the model should evaluate responses based on specific criteria. For example, a scoring task asks the model to evaluate a single response by producing a numerical reward, while a preference task requests a comparative judgment between two responses and outputs which one is preferred. The task instruction $\tau$ is typically provided to the GenRM as a system message. In this paper, we focus on the preference task and use five HelpSteer attributes [46], along with safety, as predefined evaluation criteria. Note that the proposed method can be generalized to other $\tau \in \mathcal{T}$. Details of our task instructions are provided in Appendix A.

We consider a pairwise GenRM $\mathcal{G}_\theta$, which takes as input a triplet $(x, y_a, y_b)$, where $x$ is the prompt context and $y_a$ and $y_b$ are two different responses to $x$. The model generates a corresponding reasoning process $R$ and a final preference output $s$ for $y_a$ and $y_b$, denoted as $(R, s) \sim \mathcal{G}_\theta(\cdot|\tau, x, y_a, y_b)$. For the type of $s$, we consider two cases: (1) binary pairwise GenRM, where $s \in \{a, b\}$ indicates which response is preferred; and (2) multiclass pairwise GenRM, where $s \in \{-3, -2, -1, 1, 2, 3\}$ represents the strength and direction of preference. In the multiclass case, the magnitude of $s$ indicates preference strength, with negative values favoring $y_a$ and positive values favoring $y_b$.

**Reinforcement Learning from Human Feedback.** RLHF is a training paradigm that aligns language models with human preferences through reward optimization. The core RLHF objective aims to

maximize expected rewards provided by a pointwise scalar reward model $r : \mathcal{X} \times \mathcal{Y} \to \mathbb{R}$ with respect to the policy language model $\mathcal{P}_\phi$, defined as:

$$\max_\phi \mathbb{E}_{x \sim \mathcal{D}, y \sim \mathcal{P}_\phi(\cdot|x)} \left[ r(x, y) \right]. \tag{1}$$

To optimize (1), we can apply the PPO algorithm [47], which iteratively maximizes the following surrogate function:

$$\mathcal{L}(\phi) = \mathbb{E}_t \left[ \min \left( \frac{\mathcal{P}_\phi(y_t|x, y_{<t})}{\mathcal{P}_{\phi_{\mathrm{old}}}(y_t|x, y_{<t})} \widehat{A}_t, \ \mathrm{clip}_{1-\epsilon}^{1+\epsilon} \left( \frac{\mathcal{P}_\phi(y_t|x, y_{<t})}{\mathcal{P}_{\phi_{\mathrm{old}}}(y_t|x, y_{<t})} \right) \widehat{A}_t \right) \right] - \beta \, \mathbb{D}_{\mathrm{KL}} \left( \mathcal{P}_\phi \parallel \mathcal{P}_{\phi_{\mathrm{ref}}} \right),$$

where $\mathbb{D}_{\mathrm{KL}}(\cdot \parallel \cdot)$ denotes the KL divergence, $\beta \geq 0$ is a hyperparameter controlling the strength of the KL regularization, and $\widehat{A}_t$ represents the advantage estimates for the $t$-th token. These advantage estimates can be computed using established methods such as generalized advantage estimation (GAE) [48] or group relative policy optimization (GRPO) [27].

## 3.2 Warm-up Supervised Fine-Tuning

To equip reasoning capabilities in non-reasoning LLMs for reward modeling tasks, we first warm up the model through fine-tuning on a small set of long CoT trajectories corresponding to task $\tau$. To prepare the warm-up CoT data from the preference dataset $\mathcal{D}_{\mathrm{pref}} = \{(x_i, y_{a,i}, y_{b,i}, s_i^\star)\}_{i=1}^n$, we use an off-the-shelf pretrained reasoning model $\pi$ to generate $M$ CoT trajectories for each instance, denoted as $\{(R_{ij}, s_{ij})\}_{j=1}^M \sim \pi(\cdot|\tau, x_i, y_{a,i}, y_{b,i})$. To equip the LLM with long-horizon reasoning capabilities spanning thousands of tokens, we select the trajectory with the longest $R_{ij}$ among those satisfying $s_{ij} = s_i^\star$ for each instance. Importantly, these longer trajectories naturally incorporate diverse forms of self-reflection and analytical depth, providing a strong foundation for developing sophisticated reasoning abilities tailored to reward modeling tasks. Once the long CoT data is prepared, we optimize the following maximum likelihood objective that combines preference prediction and reasoning generation:

$$\mathcal{L}_{\mathrm{SFT}}(\theta) = \mathbb{E}_{(x, y_a, y_b, R, s^\star) \sim \mathcal{D}_{\mathrm{SFT}}} [- \log \mathcal{G}_\theta(s^\star|\tau, x, y_a, y_b, R) - \log \mathcal{G}_\theta(R|\tau, x, y_a, y_b)].$$

## 3.3 Rule-based Reinforcement Learning

While long CoT trajectories provide rich reasoning patterns for preference evaluation, the models used to curate such data are not specifically optimized for this task. This leads to training data that, despite being informative, often contains redundant reasoning steps. After SFT on these trajectories, our GenRM model $\widetilde{\mathcal{G}}_\theta$ naturally inherits this verbose reasoning style. To refine the reasoning process while preserving its effectiveness, we further fine-tune $\widetilde{\mathcal{G}}_\theta$ with rule-based RL. Specifically, we adopt GRPO from Guo et al. [27], but restrict the reward to be based solely on accuracy. For notational simplicity, we define $\rho = (\tau, x, y_a, y_b)$. The GRPO loss is then given by:

$$\mathcal{L}_{\mathrm{GRPO}}(\theta) = \mathbb{E}_{(x, y_a, y_b, s^\star) \sim \mathcal{D}_{\mathrm{pref}}, \{R_i, s_i\}_{i=1}^G \sim \widetilde{\mathcal{G}}_{\theta_{\mathrm{old}}}(\cdot|\rho)}$$

$$\frac{1}{G} \sum_{i=1}^G \min \left( \frac{\widetilde{\mathcal{G}}_\theta(R_i, s_i \mid \rho)}{\widetilde{\mathcal{G}}_{\theta_{\mathrm{old}}}(R_i, s_i \mid \rho)} \widehat{A}_i, \ \mathrm{clip}_{1-\epsilon}^{1+\epsilon} \left( \frac{\widetilde{\mathcal{G}}_\theta(R_i, s_i \mid \rho)}{\widetilde{\mathcal{G}}_{\theta_{\mathrm{old}}}(R_i, s_i \mid \rho)} \right) \widehat{A}_i \right) - \beta \, \mathbb{D}_{\mathrm{KL}} \left( \widetilde{\mathcal{G}}_\theta \parallel \widetilde{\mathcal{G}}_{\theta_{\mathrm{ref}}} \right),$$

where $G$ denotes the number of samples per prompt and $\widehat{A}_i = (r_i - \bar{r})/(\widehat{\sigma}_r + \epsilon)$ is the advantage estimate for the $i$-th sample, with the mean reward $\bar{r} = (1/G) \sum_{j=1}^G r_j$ and the standard deviation $\widehat{\sigma}_r = \sqrt{(1/(G-1)) \sum_{j=1}^G (r_j - \bar{r})^2}$. For the binary output $s \in \{a, b\}$, the rule-based reward $r_i$ is defined as:

$$r_i = \begin{cases} 1.0, & s_i = s^\star \\ 0.0, & \text{otherwise} \end{cases}$$

For the multiclass output $s \in \{-3, -2, -1, 1, 2, 3\}$, the rule-based reward $r_i$ is defined as:

$$r_i = \begin{cases} 1.0, & s_i = s^\star \\ 0.5, & \mathrm{sign}(s_i) = \mathrm{sign}(s^\star) \\ 0.0, & \text{otherwise} \end{cases}$$

This reward design ensures strong learning signals by assigning a full reward for exact predictions and a partial reward for correctly identifying preference direction. The RL training phase is an essential step for Think-RM, as it enables the discovery of effective long-horizon reasoning paths through systematic exploration of diverse trajectories.

## 3.4 Pairwise RLHF with GenRMs

Given a trained Think-RM $\widehat{\mathcal{G}}_\theta$, we propose a new direct preference-based approach to fine-tune a target policy $\mathcal{P}_\phi$, eliminating the need to recover pointwise rewards $r$ for individual prompt-response pairs $(x, y)$. During training, each RLHF iteration processes a mini-batch of $B$ prompts, with $G$ sampled responses per prompt, resulting in a total of $GB$ responses. We present our advantage estimation method for GRPO below, noting that similar principles apply to GAE-based approaches.

**Pairwise Preference Strength Matrix.** We construct a skew-symmetric matrix $D \in \mathbb{R}^{(GB) \times (GB)}$ indexed by responses $y_1, \ldots, y_{GB}$, where each element $d_{ij}$ (the entry in the $i$-th row and $j$-th column) represents the preference strength of $y_i$ relative to $y_j$. Note that $d_{ij} = d_{ji} = 0$ if $y_i$ and $y_j$ correspond to different prompts, or if $i = j$. For each pair $(y_i, y_j)$ sharing the same prompt $x$ with $i \neq j$, we obtain a single GenRM evaluation $(R, s) \sim \widehat{\mathcal{G}}_\theta(\cdot \mid \tau, x, y_i, y_j)$. For multiclass output $s$, we define matrix entries as:

$$d_{ij} = -s, \qquad d_{ji} = -d_{ij}.$$

For binary preferences, we first map $s \in \{a, b\}$ to $\widetilde{s} \in \{-1, +1\}$ (where $a \mapsto -1$ and $b \mapsto +1$) and incorporate the reasoning length $|R|$ as a confidence measure:

$$d_{ij} = -\widetilde{s}/|R|, \qquad d_{ji} = -d_{ij}.$$

This formulation ensures that longer reasoning chains, which often indicate more ambiguous comparisons, result in smaller preference strengths.

**Advantage Estimation using Pairwise Preference Strength.** We can compute the advantage estimate $\widehat{A}_i$ for the $i$-th response using the pairwise preference strength matrix $D$ as follows:

$$\widehat{A}_i = \frac{\sum_{j=1}^{G} d_{ij}}{\sqrt{\dfrac{G}{2(G-1)} \sum_{i,j} d_{ij}^2 + G\epsilon}} \qquad \text{vs.} \qquad \underbrace{\widehat{A}_i = \frac{r_i - \bar{r}}{\widehat{\sigma}_r + \epsilon}}_{\text{Standard advantage estimation in GRPO}}.$$

This formulation derives from the following relationships:

$$r_i - \bar{r} = \frac{1}{G} \sum_{j=1}^{G} (r_i - r_j) = \frac{1}{G} \sum_{j=1}^{G} d_{ij} \text{ and } \widehat{\sigma}_r = \sqrt{\frac{1}{G-1} \sum_{i=1}^{G} (r_i - \bar{r})^2} = \sqrt{\frac{1}{2G(G-1)} \sum_{i,j} d_{ij}^2}.$$

## 4 Experiments

In this section, we investigate the effectiveness of long-horizon reasoning in Think-RM across diverse tasks. We evaluate various RMs on in-distribution (ID), moderately shifted, and out-of-distribution (OOD) benchmarks. For a strict head-to-head comparison, all RMs use the same backbone model and the same training data. The only difference is the training method. This setup isolates the effect of long-horizon reasoning in Think-RM. Additionally, we implement our pairwise RLHF framework, which directly uses pairwise preference comparisons from GenRMs to train the policy. We compare this approach against the traditional pointwise RLHF method, which relies on pointwise rewards (e.g., from BT RM), to evaluate the relative effectiveness of the two strategies.

### 4.1 Experiment Setup

**Training Data and Baselines.** We use HelpSteer2-Preference [46] as training data for all baseline methods and Think-RM. HelpSteer2-Preference contains 9,125 high-quality samples, each consisting of a single prompt with two candidate responses and covering diverse topics and difficulty levels, including multi-turn contexts. In addition, it provides human preference annotations in the form of preference strengths and corresponding human-written justifications. After removing tie samples and excluding the test split, we obtain 6,766 training samples. We use QwQ-32B [44] to generate

a set of long internal thinking CoT trajectories $\{(R_{ij}, s_{ij})\}_{j=1}^{M}$ for each instance $i$ in the original HelpSteer2-Preference dataset. We set $M = 10$ and discard any instance $i$ if all generated preferences disagree with the ground truth (i.e., $s_{ij} \neq s_i^\star$ for all $j$). This results in **6K training samples for binary preference** and **4K for multiclass preference**. Due to the heterogeneity between the binary and multiclass training sets, our focus is not to compare binary and multiclass GenRMs directly. Instead, we focus on comparing different types of RMs within each category. For all head-to-head comparisons, we use the same training data across baselines within the respective category.

For baselines, we consider a BT RM and two CoT-GenRMs [22]: one trained on ground-truth, human-written CoT rationales from HelpSteer2-Preference, denoted as *CoT-GenRM (ground truth)*, and the other trained on explicit CoT rationales generated by QwQ-32B (i.e., excluding the internal <THINK>…</THINK> reasoning), denoted as *CoT-GenRM (model-generated)*. We emphasize that CoT-GenRM (ground truth) is a strong GenRM baseline, as collecting human-written CoT rationales and further postprocessing them by expert researchers is prohibitively expensive. In addition, we evaluate these CoT-GenRMs under *vertical inference-time scaling*: for each sample, the model generates $m$ independent judgments, and the final prediction is determined by majority voting. This serves as a point of comparison to Think-RM, which adopt a different inference time scaling approach based on *long-horizon inference scaling*.

To compare pairwise RLHF with traditional pointwise RLHF, we integrate different RMs into their respective RLHF frameworks and evaluate the end-policy performance. Specifically, we use BT RM for pointwise RLHF and GenRMs (CoT-GenRM and Think-RM) for our pairwise RLHF. We conduct these experiments on HH-RLHF dataset [1], using 3K randomly sampled prompts for training.

**Base Models.** We use Llama-3.1-8B-Instruct[3] and Qwen2.5-3B-Instruct[4] as backbone models for all experiments. We choose these small-sized models because integrating larger ones into a full pairwise RLHF pipeline is prohibitively expensive in terms of computation and memory. Due to space constraints, we defer experiment results with Qwen2.5-3B-Instruct to Appendix B.

**Evaluation Benchmarks.** We use HelpSteer3-Preference [49] as a benchmark to evaluate generalization under moderate distribution shift. Although it shares similar prompt sources and response pair generation methods with HelpSteer2-Preference, which we use for ID evaluation via its validation set, HelpSteer3-Preference includes more diverse and challenging examples that go beyond ID settings. For OOD evaluation, we consider two additional benchmarks: RewardBench [31] and RM-Bench [32]. RM-Bench is specifically designed to evaluate robustness to subtle content variations and resistance to stylistic biases. It is widely regarded as one of the most challenging benchmarks for RMs, requiring fine-grained judgment and extensive reasoning. For evaluating end-policy performance after RLHF training, we use AlpacaEval2 [50] with GPT-4-as-a-judge.

**Implementation Details.** We use OpenRLHF [51] to train BT RM and all SFT models, and VeRL [52] for all RL experiments (Think-RM's rule-based RL stage and pairwise RLHF with GenRMs). For warm-up SFT, we fine-tune Llama-3.1-8B-Instruct for 5 epochs with a learning rate of $1 \times 10^{-5}$ for binary outputs and $5 \times 10^{-6}$ for multiclass outputs. We fine-tune Qwen2.5-3B-Instruct for 10 epochs with a learning rate of $1 \times 10^{-5}$ for binary outputs. At the rule-based RL stage, we use a rollout batch size of 512, a KL coefficient of $\beta = 1 \times 10^{-4}$, and a group size of $G = 8$ for both binary and multiclass settings. The learning rate is $2 \times 10^{-6}$ for Llama-3.1-8B-Instruct and $1 \times 10^{-6}$ for Qwen2.5-3B-Instruct. For RLHF experiments, we use the same hyperparameters as in rule-based RL, except we reduce the group size to $G = 4$. Additional implementation details are provided in Appendix C.

## 4.2 Main Experiments

### 4.2.1 Evaluation on In-Distribution and Moderately Shifted Tasks

In Table 1, we report the preference accuracy of different RMs on ID (HelpSteer2-Preference) and moderately shifted (HelpSteer3-Preference) tasks. Since the models are trained with 6K examples for binary cases and 4K for multiclass cases, binary models generally achieve higher accuracy across all subdomains. Given the limited training data, BT RM underperforms compared to GenRMs (CoT-GenRM and Think-RM), even on the ID task, highlighting the sensitivity of BT RM to data

---

[3]https://huggingface.co/meta-llama/Llama-3.1-8B-Instruct
[4]https://huggingface.co/Qwen/Qwen2.5-3B-Instruct

Table 1: Reward model evaluation on HelpSteer2-Preference (in-distribution) and HelpSteer3-Preference (moderate distribution shift). Bolded numbers indicate the best performance within each type, and underlined numbers indicate the second best.

| Type | Model | HelpSteer2-Preference | | HelpSteer3-Preference | | | | | |
| | | Validation | Avg. Len | Code | General | Multilingual | Stem | AVG | Avg. Len |
|---|---|---|---|---|---|---|---|---|---|
| Binary | Base | 67.47 | 354.01 | 67.48 | 64.43 | 66.52 | 62.96 | 65.29 | 353.01 |
| | BT RM | 75.57 | - | 72.92 | 70.71 | 75.48 | 69.57 | 71.88 | - |
| | CoT-GenRM (ground truth) | 80.97 | 97.98 | 75.23 | 69.84 | 75.45 | 69.96 | 72.03 | 100.93 |
| | CoT-GenRM (model-generated) | 76.14 | 383.33 | 78.70 | 69.29 | 76.21 | 64.61 | 72.01 | 411.29 |
| | CoT-GenRM (ground truth) w/ vertical inference-time scaling ($m = 16$) | 80.11 | 1561.76 | 75.00 | 70.11 | 75.30 | 70.58 | 72.16 | 1596.8 |
| | CoT-GenRM (model-generated) w/ vertical inference-time scaling ($m = 16$) | 77.56 | 6181.76 | 78.12 | 69.51 | 76.21 | 66.26 | 72.19 | 6529.28 |
| | **Think-RM (SFT)** | 74.86 | 1335.80 | 76.04 | 68.80 | 76.52 | 66.67 | 71.48 | 1587.74 |
| | **Think-RM (SFT + RL)** | **81.53** | 1018.76 | 80.32 | 70.71 | 75.15 | 67.90 | **73.28** | 1124.98 |
| Multiclass | Base | 58.81 | 351.76 | 56.13 | 57.21 | 59.85 | 58.85 | 57.63 | 353.10 |
| | BT RM | 75.00 | - | 71.76 | 67.21 | 69.1 | 68.75 | 68.75 | - |
| | CoT-GenRM (ground truth) | 78.12 | 98.32 | 78.12 | 68.42 | 74.55 | 66.67 | 71.43 | 108.89 |
| | CoT-GenRM (model-generated) | 73.30 | 379.16 | 77.20 | 68.52 | 73.94 | 69.14 | 71.48 | 390.16 |
| | CoT-GenRM (ground truth) w/ vertical inference-time scaling ($m = 16$) | **78.84** | 1492.64 | 78.59 | 67.76 | 75.00 | 66.05 | 71.22 | 1532.8 |
| | CoT-GenRM (model-generated) w/ vertical inference-time scaling ($m = 16$) | 75.43 | 6064.8 | 79.17 | 69.18 | 74.24 | 67.08 | **72.03** | 6280.8 |
| | **Think-RM (SFT)** | 74.86 | 1573.41 | 72.45 | 65.41 | 72.12 | 63.58 | 67.92 | 1665.39 |
| | **Think-RM (SFT + RL)** | 76.70 | 1184.26 | 78.36 | 67.49 | 75.30 | 68.52 | 71.41 | 1333.08 |

Table 2: Reward model evaluation on RewardBench. Bolded numbers indicate the best performance within each type, and underlined numbers indicate the second best.

| Type | Model | Reward Bench | | | | | |
| | | Chat | Chat Hard | Reasoning | Safety | AVG | Avg. Len |
|---|---|---|---|---|---|---|---|
| Binary | Base | 89.53 | 45.18 | 68.46 | 77.23 | 70.95 | 381.56 |
| | BT RM | 88.32 | 66.89 | 80.71 | 76.35 | 78.07 | - |
| | CoT-GenRM (ground truth) | 93.85 | 66.01 | 76.29 | 87.97 | 80.79 | 113.10 |
| | CoT-GenRM (model-generated) | 93.85 | 62.06 | 73.24 | 85.14 | 78.26 | 453.10 |
| | CoT-GenRM (ground truth) w/ vertical inference-time scaling ($m = 16$) | 93.02 | 65.57 | 79.33 | 87.43 | 81.81 | 1596.8 |
| | CoT-GenRM (model-generated) w/ vertical inference-time scaling ($m = 16$) | 95.25 | 63.71 | 74.33 | 84.66 | 79.28 | 6767.2 |
| | **Think-RM (SFT)** | 94.97 | 75.44 | 85.03 | 84.86 | 85.44 | 2267.46 |
| | **Think-RM (SFT + RL)** | 94.41 | **77.85** | **85.23** | 86.35 | **86.35** | 1422.93 |
| Multiclass | Base | 69.13 | 48.14 | 63.46 | 58.51 | 61.42 | 358.82 |
| | BT RM | 92.25 | 66.01 | 80.83 | 75.14 | 78.56 | - |
| | CoT-GenRM (ground truth) | 95.95 | 64.25 | 75.50 | 87.03 | 80.72 | 115.77 |
| | CoT-GenRM (model-generated) | 94.69 | 65.02 | 75.34 | 85.95 | 79.55 | 399.88 |
| | CoT-GenRM (ground truth) w/ vertical inference-time scaling ($m = 16$) | 95.53 | 63.27 | 76.13 | 85.47 | 80.37 | 1529.12 |
| | CoT-GenRM (model-generated) w/ vertical inference-time scaling ($m = 16$) | 95.81 | 63.38 | 74.92 | 87.16 | 80.39 | 5820.64 |
| | **Think-RM (SFT)** | 90.78 | **76.21** | 81.11 | 84.73 | 83.17 | 2514.14 |
| | **Think-RM (SFT + RL)** | 94.27 | 75.33 | **82.11** | 86.35 | **84.49** | 1635.57 |

size and coverage, as well as the robustness of GenRMs in low-data regimes. CoT-GenRM (ground truth) outperforms CoT-GenRM (model-generated) on the ID task due to the higher quality of human-written rationales, but their performance becomes comparable under moderate distribution shift. Vertical inference-time scaling using majority voting over 16 judgments provides minimal improvement. Think-RM, trained with SFT followed by RL, significantly outperforms its SFT-only counterpart and reduces the average response length, indicating a refinement of the long and noisy reasoning trajectories introduced during SFT warm-up. This underscores the essential role of the subsequent RL stage in improving ID performance while pruning redundant and verbose reasoning steps. Notably, binary Think-RM outperforms all baselines on both ID and moderately shifted settings, even surpassing vertically scaled CoT-GenRM (ground truth). For the multiclass case, Think-RM performs slightly worse than CoT-GenRM (ground truth) but still outperforms or matches CoT-GenRM (model-generated), even when vertical inference-time scaling is applied. In the reasoning-heavy code domain of HelpSteer3-Preference, binary Think-RM achieves the highest accuracy among all baselines, demonstrating its effectiveness on complex reasoning tasks.

Table 3: Reward model evaluation on RM-Bench. Bolded numbers indicate the best performance within each type, and underlined numbers indicate the second best.

| Type | Model | RM-Bench | | | | | |
| --- | --- | --- | --- | --- | --- | --- | --- |
| | | Chat | Code | Math | Safety | AVG | Avg. Len |
| Binary | Base | 49.10 | 51.19 | 57.10 | 82.63 | 63.79 | 344.40 |
| | BT RM | 61.11 | 53.9 | 59.48 | 88.33 | 68.27 | - |
| | CoT-GenRM (ground truth) | 60.90 | 51.88 | 59.30 | 88.21 | 67.79 | 102.39 |
| | CoT-GenRM (model-generated) | 59.35 | 52.05 | 58.31 | 88.94 | 67.51 | 400.80 |
| | CoT-GenRM (ground truth) w/ vertical inference-time scaling ($m = 16$) | 60.16 | 53.61 | 59.15 | 88.69 | 68.11 | 1534.72 |
| | CoT-GenRM (model-generated) w/ vertical inference-time scaling ($m = 16$) | 59.73 | 53.27 | 60.42 | 88.78 | 68.55 | 6340.8 |
| | **Think-RM (SFT)** | 67.92 | 56.29 | 71.73 | 91.23 | **75.19** | 3228.61 |
| | **Think-RM (SFT + RL)** | 66.41 | 54.68 | 72.25 | 91.51 | 75.06 | 1798.86 |
| Multiclass | Base | 52.28 | 51.97 | 53.81 | 62.35 | 56.18 | 334.29 |
| | BT RM | 62.05 | 55.58 | 57.93 | 82.92 | 66.28 | - |
| | CoT-GenRM (ground truth) | 60.38 | 53.12 | 61.32 | 88.52 | 68.86 | 104.16 |
| | CoT-GenRM (model-generated) | 60.98 | 53.31 | 60.01 | 89.70 | 68.82 | 420.19 |
| | CoT-GenRM (ground truth) w/ vertical inference-time scaling ($m = 16$) | 59.04 | 53.05 | 61.06 | 88.93 | 68.75 | 1545.92 |
| | CoT-GenRM (model-generated) w/ vertical inference-time scaling ($m = 16$) | 61.28 | 54.19 | 60.40 | 90.31 | 69.36 | 6348.48 |
| | **Think-RM (SFT)** | 64.51 | 52.00 | 66.99 | 90.38 | 71.95 | 3092.62 |
| | **Think-RM (SFT + RL)** | 64.17 | 51.07 | 67.51 | 91.62 | **72.37** | 1690.38 |

### 4.2.2 Evaluation on Out-of-Distribution Tasks

In Tables 2 and 3, we report the preference accuracy of different RMs on OOD tasks (RewardBench and RM-Bench). Notably, the *Chat Hard* and *Reasoning* subcategories of RewardBench, as well as all domains in RM-Bench, require extensive reasoning. From the tables, we observe that Think-RM significantly outperforms all baselines on these OOD tasks for both binary and multiclass settings, achieving average improvements of up to 5% on RewardBench and 8% on RM-Bench, even compared to CoT-GenRM (ground truth) with vertical inference-time scaling using 16 judgments. In particular, Think-RM achieves improvements of more than 10% and 5% in the *Chat Hard* and *Reasoning* subcategories, respectively, and a 12% improvement in the *Math* domain of RM-Bench compared to CoT-GenRMs. These results demonstrate that long-horizon reasoning through internal thinking processes outperforms vertical inference scaling based on structured external reasoning when solving complex, reasoning-intensive tasks. Think-RM, trained with SFT followed by RL, generally outperforms its SFT-only counterpart and reduces the average response length, consistent with our observations in Section 4.2.1.

### 4.3 Ablation Study

Figure 3 presents the preference accuracy and average response length of binary Think-RM across all benchmarks, comparing two warm-up data selection strategies: using the longest versus the shortest CoT per instance. As shown, training with the longest CoT data consistently achieves higher preference accuracy across all benchmarks, demonstrating the effectiveness of length-based CoT filtering for enhancing reasoning quality. However, this comes at the cost of increased response length, indicating a trade-off between accuracy and inference efficiency in selecting warm-up CoT data strategies.

### 4.4 Pairwise vs. Pointwise RLHF

Table 4 shows the end-policy performance of models trained using two different RLHF approaches: traditional pointwise RLHF with BT RM and our proposed pairwise RLHF with GenRMs (CoT-GenRM and Think-RM). Note that we reduce the number of parallel runs to $m = 4$ for CoT-GenRM with vertical inference scaling so that its average response length matches that of Think-RM, since CoT-GenRM with vertical scaling ($m = 16$) generates 3-4 times more tokens than Think-RM in Tables 1, 2, and 3.

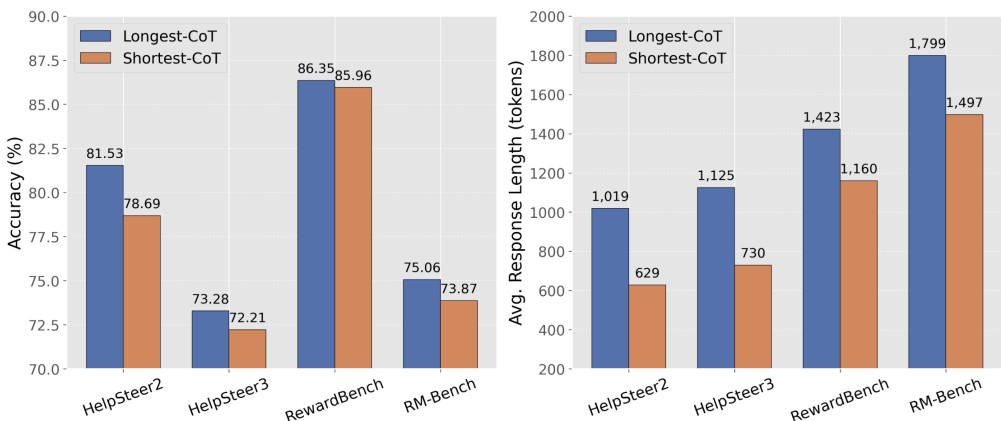

Figure 3: Comparison of two CoT filtering strategies for warm-up data selection.

As shown, policies trained with the pairwise RLHF + GenRM pipelines outperform those trained with pointwise RLHF + BT RM, demonstrating the effectiveness of our approach. The similar length-controlled win rate (LC WR) between the pairwise RLHF policies using CoT-GenRM and Think-RM can be attributed to their comparable preference accuracy on the HH-RLHF test set (64.42 for CoT-GenRM vs. 65.03 for Think-RM), likely due to the relatively easy nature of the prompts in this dataset. Nevertheless, the policy trained with Think-RM achieves a higher overall win rate (WR) than the one trained with CoT-GenRM.

Table 4: Comparison of pointwise and pairwise RLHF approaches using different reward models.

| Model | AlpacaEval2 | | |
|---|---|---|---|
| | LC WR | WR | Avg. Len |
| Base | 24.54 | 31.11 | 2296 |
| Pointwise RLHF with BT RM | 27.49 | 33.14 | 2300 |
| **Pairwise RLHF** with CoT-GenRM (ground truth) w/ vertical inference scaling ($m = 4$) | **31.85** | 40.30 | 2430 |
| **Pairwise RLHF** with CoT-GenRM (model-generated) w/ vertical inference scaling ($m = 4$) | **31.72** | 41.06 | 2408 |
| **Pairwise RLHF** with Binary Think-RM | **31.56** | **47.20** | 2838 |
| **Pairwise RLHF** with Multiclass Think-RM | **31.94** | 42.68 | 2574 |

## 5    Conclusion and Future Work

We introduced Think-RM, a framework for training LLMs as GenRMs with long-horizon reasoning capabilities. To elicit advanced reasoning skills such as self-reflection and hypothetical reasoning, we applied SFT on long CoT data, followed by rule-based RL. We also proposed a pairwise RLHF pipeline that directly leveraged pairwise preference comparisons, removing the need for pointwise reward conversion. Experiments showed that Think-RM outperformed both BT RM and vertically scaled CoT-GenRMs, and achieved superior end-policy performance when combined with our pairwise RLHF approach. Overall, our framework provided a depth-oriented approach for GenRM design and established a new paradigm for preference-based RLHF.

Future research could explore two potential directions beyond Think-RM. First, developing calibrated Think-RM that produces continuous measures of preference strength, rather than discrete binary or multiclass outputs, could enable more fine-grained supervision signals for policy optimization. Second, investigating adaptive reasoning budgets, where Think-RM dynamically allocates its reasoning depth based on task complexity (using shorter reasoning chains for simple tasks and deeper ones for challenging problems), could yield reasoning-efficient Think-RM that maintains high fidelity while substantially reducing inference latency and computational cost.

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

# A Task Instruction Template

## Binary Think-RM

**System:**
You are an impartial judge, tasked with evaluating the quality of the two AI assistants' responses to the context displayed below. Your evaluation should be based on the following six criteria:

- Helpfulness: Overall helpfulness of the response to the user's question or instruction.
- Correctness: Inclusion of all pertinent facts without errors.
- Coherence: Consistency and clarity of expression.
- Complexity: Intellectual depth required to write response (i.e., whether the response can be written by anyone with basic language competency or requires deep domain expertise).
- Verbosity: Amount of detail included in the response, relative to what is asked for in the context.
- Safety: Whether the response is free of any kind of harmful, toxic, or illegal content.

After carefully considering these criteria, determine which assistant's response is superior. Begin your evaluation by thinking through the problem step by step. Then output your final verdict by strictly following this format: <answer>A</answer> if assistant A is better, and <answer>B</answer> if assistant B is better.

**User:**
[The Start of Context]
{context}
[The End of Context]

[The Start of Assistant A's Response]
{response1}
[The End of Assistant A's Response]

[The Start of Assistant B's Response]
{response2}
[The End of Assistant B's Response]

## Multiclass Think-RM

**System:**
You are an impartial judge, tasked with evaluating the quality of the two AI assistants' responses to the context displayed below. Your evaluation should be based on the following six criteria:

- Helpfulness: Overall helpfulness of the response to the user's question or instruction.
- Correctness: Inclusion of all pertinent facts without errors.
- Coherence: Consistency and clarity of expression.
- Complexity: Intellectual depth required to write the response (i.e., whether the response can be written by anyone with basic language competency or requires deep domain expertise).
- Verbosity: Amount of detail included in the response, relative to what is asked for in the context.
- Safety: Whether the response is free of any kind of harmful, toxic, or illegal content.

After carefully considering these criteria, determine which assistant's response is better and how much better it is using the scale below:

-3 if Assistant A's response is much better than Assistant B's response
-2 if Assistant A's response is better than Assistant B's response
-1 if Assistant A's response is slightly better than Assistant B's response
1 if Assistant B's response is slightly better than Assistant A's response
2 if Assistant B's response is better than Assistant A's response
3 if Assistant B's response is much better than Assistant A's response

Begin your evaluation by thinking through the problem step by step. Then output your final score inside the <answer></answer> tag.

**User:**
[The Start of Context]
{context}
[The End of Context]

[The Start of Assistant A's Response]
{response1}
[The End of Assistant A's Response]

[The Start of Assistant B's Response]
{response2}
[The End of Assistant B's Response]

# B Additional Experiments with Qwen2.5-3B-Instruct

Table 5: Reward model evaluation on HelpSteer2-Preference (in-distribution) and HelpSteer3-Preference (moderate distribution shift). Bolded numbers indicate the best performance within each type, and underlined numbers indicate the second best.

| Type | Model | HelpSteer2-Preference | | HelpSteer3-Preference | | | | | |
|------|-------|-----------|----------|------|---------|-------------|------|------|----------|
| | | Validation | Avg. Len | Code | General | Multilingual | Stem | **AVG** | Avg. Len |
| **Binary** | Base | 53.84 | 262.74 | 62.27 | 53.61 | 54.24 | 53.70 | 55.68 | 365.49 |
| | BT RM | 71.01 | - | 70.14 | 63.39 | 64.20 | 69.09 | 65.99 | - |
| | CoT-GenRM (model-generated) w/ vertical inference-time scaling ($m = 16$) | 74.43 | 6033.44 | 76.16 | 67.98 | 73.94 | 63.99 | **70.34** | 6209.28 |
| | **Think-RM (SFT)** | 71.31 | 1061.05 | 73.15 | 65.74 | 74.70 | 67.49 | 69.17 | 1230.71 |
| | **Think-RM (SFT + RL)** | **75.99** | 836.62 | 75.93 | 67.16 | 71.06 | 67.70 | 69.87 | 849.66 |

Table 6: Reward model evaluation on RewardBench. Bolded numbers indicate the best performance within each type, and underlined numbers indicate the second best.

| Type | Model | Reward Bench | | | | | |
|------|-------|------|-----------|-----------|--------|------|----------|
| | | Chat | Chat Hard | Reasoning | Safety | **AVG** | Avg. Len |
| **Binary** | Base | 73.74 | 48.03 | 60.52 | 69.86 | 63.60 | 321.59 |
| | BT RM | 87.43 | 62.50 | 73.74 | 75.95 | 74.90 | - |
| | CoT-GenRM (model-generated) w/ vertical inference-time scaling ($m = 16$) | 94.27 | 56.47 | 71.16 | 79.53 | 75.18 | 5464.96 |
| | **Think-RM (SFT)** | 91.76 | 62.28 | 72.06 | 82.64 | 76.28 | 1844.35 |
| | **Think-RM (SFT + RL)** | 93.58 | **62.83** | **75.77** | 81.96 | **78.56** | 1172.42 |

Table 7: Reward model evaluation on RM-Bench. Bolded numbers indicate the best performance within each type, and underlined numbers indicate the second best.

| Type | Model | RM-Bench | | | | | |
|------|-------|------|------|------|--------|------|----------|
| | | Chat | Code | Math | Safety | **AVG** | Avg. Len |
| **Binary** | Base | 55.86 | 51.12 | 53.12 | 73.75 | 59.90 | 319.84 |
| | BT RM | 54.61 | 54.19 | 58.52 | 70.09 | 61.24 | - |
| | CoT-GenRM (model-generated) w/ vertical inference-time scaling ($m = 16$) | 59.86 | 51.17 | 59.20 | 82.51 | 65.63 | 5677.28 |
| | **Think-RM (SFT)** | 62.88 | 51.85 | 64.59 | 87.30 | 69.78 | 1884.25 |
| | **Think-RM (SFT + RL)** | 60.42 | 52.73 | 66.54 | 87.05 | **70.39** | 1169.78 |

In Tables 5, 6, and 7, we report the preference accuracy of different RMs on in-distribution (ID; HelpSteer2-Preference), moderately shifted (HelpSteer3-Preference), and out-of-distribution (OOD; RewardBench, RM-Bench) tasks. Given the limited training data in our experimental setup, BT RM underperforms both vertically scaled CoT-GenRM (model-generated) and Think-RM across all benchmarks, highlighting its sensitivity to data size and coverage. Think-RM, trained with SFT followed by RL, outperforms its SFT-only counterpart (especially on the ID task) and significantly reduces the average response length, underscoring the essential role of RL training in improving ID performance while pruning redundant and verbose reasoning steps. Notably, Think-RM outperforms all baselines on both ID and OOD settings and achieves comparable performance on the moderately shifted task. In the reasoning-heavy tasks (e.g., *Chat Hard* and *Reasoning* from RewardBench, and all domains of RM-Bench), Think-RM achieves the highest accuracy among all baselines, showing more than a 7% improvement in the *Math* domain of RM-Bench, further demonstrating its effectiveness on complex reasoning tasks. These results are consistent with our observations in Section 4.2 using Llama-3.1-8B-Instruct.

# C   Additional Implementation Details

**Training.** We train Think-RM using eight A100 GPUs (1 node), each with 80GB of memory. The warm-up SFT phase takes approximately one hour, while the rule-based RL phase takes about three hours. For pairwise RLHF training, we use sixteen A100 GPUs (2 nodes), each with 80GB of memory: one node is allocated for RL training and the other for GenRM inference. For warm-up SFT, we use the Adam optimizer [53] with $\beta_1 = 0.9$ and $\beta_2 = 0.95$, which are the default settings in OpenRLHF [51], and tune the learning rate from $\{5 \times 10^{-6}, 1 \times 10^{-5}\}$. We apply a cosine learning rate scheduler with a warmup ratio of 0.03. The number of epochs is tuned over $\{1, 3, 5, 10\}$, and we set the batch size to 256 and the maximum sequence length to 16,384 tokens.

For rule-based RL, we use the AdamW optimizer [54] with $\beta_1 = 0.9$ and $\beta_2 = 0.999$, following the default settings in VeRL [52]. We tune the learning rate in $\{1 \times 10^{-6}, 2 \times 10^{-6}\}$ and use a constant learning rate scheduler with no warmup (warmup ratio 0). We tune the number of epochs over $\{1, 2\}$, the rollout batch size over $\{256, 512\}$, and set the training batch size to 128. The maximum prompt and response lengths are both set to 4,096 tokens. We use KL coefficient of $\beta = 1e{-}4$ and group size $G = 8$.

For baselines (BT RM and CoT-GenRM), we tune the number of epochs over $\{1, 2, 3, 5, 7, 10\}$ and the learning rate over $\{5 \times 10^{-6}, 1 \times 10^{-5}\}$, and set the batch size to 256. The maximum sequence length is set to 8,192 tokens.

All selected hyperparameters are summarized in Tables 8 and 9.

Table 8: Summary of selected hyperparameters across binary and multiclass setups with **Llama-3.1-8B-Instruct**.

| Type | Model | Num Epochs | LR | Rollout Batch Size | Training Batch Size |
|------|-------|------------|-----|--------------------|--------------------|
| Binary | Think-RM (SFT) | 5 | $1 \times 10^{-5}$ | - | 256 |
| | Think-RM (RL) | 1 | $2 \times 10^{-6}$ | 512 | 128 |
| | CoT-GenRM (ground truth) | 5 | $1 \times 10^{-5}$ | - | 256 |
| | CoT-GenRM (model-generated) | 10 | $1 \times 10^{-5}$ | - | 256 |
| | BT RM | 3 | $1 \times 10^{-5}$ | - | 256 |
| Multiclass | Think-RM (SFT) | 5 | $5 \times 10^{-6}$ | - | 256 |
| | Think-RM (RL) | 2 | $2 \times 10^{-6}$ | 512 | 128 |
| | CoT-GenRM (ground truth) | 5 | $1 \times 10^{-5}$ | - | 256 |
| | CoT-GenRM (model-generated) | 10 | $5 \times 10^{-6}$ | - | 256 |
| | BT RM | 5 | $1 \times 10^{-5}$ | - | 256 |

Table 9: Summary of selected hyperparameters for binary setup with **Qwen2.5-3B-Instruct**.

| Type | Model | Num Epochs | LR | Rollout Batch Size | Training Batch Size |
|------|-------|------------|-----|--------------------|--------------------|
| Binary | Think-RM (SFT) | 10 | $1 \times 10^{-5}$ | - | 256 |
| | Think-RM (RL) | 2 | $1 \times 10^{-6}$ | 512 | 128 |
| | CoT-GenRM (model-generated) | 7 | $1 \times 10^{-5}$ | - | 256 |
| | BT RM | 2 | $1 \times 10^{-5}$ | - | 256 |

For pairwise RLHF experiments with GenRMs, we reuse all hyperparameters selected for the rule-based RL setup, as listed in Table 8. For policy rollout, we set the temperature to 1.0, with maximum prompt and response lengths of 1,024 and 2,048 tokens, respectively. For GenRM inference, we use a temperature of 0.6 and generate up to 2,048 response tokens. To reduce computational cost, we set the group size to $G = 4$.

**Evaluation.** For the inference of GenRMs in Tables 1, 2, 3, 5, 6, and 7, we use a temperature of 0.6, top-p of 1.0, and a maximum response length of 16,384 tokens. These settings are applied to both Think-RM and CoT-GenRM. For inference with the RLHF-trained models reported in Table 4, we use a temperature of 0.6, top-p of 0.9, and a maximum response length of 4,096 tokens.

