# OpenReview forum: "Think-RM: Enabling Long-Horizon Reasoning in Generative Reward Models"
_NeurIPS.cc/2025/Conference — NeurIPS 2025 poster_

### Official Review · Reviewer_KhzV · 2025-06-28

**Clarity:** 1
**Significance:** 2
**Originality:** 2
**Rating:** 5
**Confidence:** 2

**Summary:**

The authors present Think-RM, a recepie for training long-horizon reasoning reward models. While LLM-as-a-judge paradigm as gained popularity, the authors point out that the standard is vertical reasoning, where many reasoning trajectories are generated and aggregated. Instead, the authors propose horizontal reasoning,g which encourages reasoning within a single trajectory. In standard Gen-RM, a dataset of reasoning traces is generated from a preference dataset. Then, a DPO-style supervised learning objective is used to train the reward model to combine preference predictions and reasoning traces. In Think-RM, the authors select the trajectory with the longest reasoning trace (and correct preference prediction) to build the training data.  The authors then apply GRPO to eliminate the model's tendency for verbose reasoning traces. Finally, the authors build a skew-matrix that down-weights the relative differences between long reasoning traces for the GRPO advantage term.

The authors empirically show that their reward models outperform standard baselines on a variety of reward model benchmarks, and also show that the LM policies learned by using their Think-RM reward models outperform baselines on the AlpacaEval2 benchmark.

**Questions:**

Do we always get long reasoning chains even if they are not required with Think-RM? Are there potentially other heuristics we could use instead of simply the longest chains to generate the training data?

I wasn't clear on why we need the skew-matrix? The authors say, "This formulation ensures that longer reasoning chains...result in smaller reward differences". I am unsure why we want this? Some clarification would be helpful.

**Ethical Concerns:**

["NO or VERY MINOR ethics concerns only"]

**Final Justification:**

I was initially sceptical that selecting the longest chains of thought would be the best heuristic; however, the authors addressed these concerns by clarifying the use of the rule-based rewards. They also outlined that their experiments actually produced shorter reasoning traces than the baselines. Finally, I appreciate that the authors added limitations to the discussion.

**Limitations:**

The authors do not discuss _any_ limitations. I would like to invite the authors to discuss some limitations of their method. For example, do we _always_ want long reasoning chains? I would think for simple questions we would not, but for long, difficult questions we do. Is there a way to get the best of both?

**Quality:**

3

**Strengths And Weaknesses:**

# Strengths
-The paper empirically shows that Think-RM outperforms COT-Gen RM reward models on RM-Bench and Reward Bench.
- The paper empirically shows that reasoning models trained using the Think-RM reward model outperform pointwise RLHF models on AlpacaEval 2, and outperform reasoning models using CoT-GenRM when the length is uncontrolled.
- The paper provides ablations showing that selecting the longest trajectory to train Think-RM is beneficial over selecting the shortest trajectory.

# Weaknesses
- I was initially sceptical that selecting the longest chain of thought would be a good idea, but the authors did partially address my concern by using rule-based RL to remove the verbose reasoning style. However, I am unsure how the rules were applied, what the rules were and how they eliminate verbosity. The authors say they "restrict the rewards to be based solely on accuracy", but I am unsure what that means exactly.
- While the paper does present a new way of training a reward model, it is not original in the sense that it remixes well-known existing training techniques.
- There is no discussion of limitations.
- I find many of the algorithmic decisions, like using CoT length and the skew matrix, unmotivated.

---

> ### Author Rebuttal · Authors · 2025-07-31
>
> > W1. I was initially sceptical that selecting the longest chain of thought would be a good idea, but the authors did partially address my concern by using rule-based RL to remove the verbose reasoning style. However, I am unsure how the rules were applied, what the rules were and how they eliminate verbosity. The authors say they "restrict the rewards to be based solely on accuracy", but I am unsure what that means exactly.
>
> In Lines 187–188 of the manuscript, we define the rule-based reward functions used for both binary and multiclass Think-RM. For binary Think-RM, we assign a reward of 1 to all tokens in the response if the model correctly predicts the preferred answer; otherwise, the reward is 0. For multiclass Think-RM, we add an additional partial reward when the model correctly identifies the preference direction, even if the final prediction is not fully correct.
>
> By applying RL with these rule-based rewards, the model explores and learns to retain the parts of reasoning essential for accuracy while eliminating verbose or redundant steps. **This outcome is evident in Tables 1–3: Think-RM (SFT + RL) outperforms Think-RM (SFT), while also producing significantly shorter internal reasoning traces.** This is also consistent with results obtained using different backbone models (Qwen2.5-3B-Instruct), as summarized in our response to Reviewer BTue.
>
> Moreover, in Section 4.3 (Ablation Study), we compare two CoT data selection strategies: using the longest vs. the shortest reasoning trace per instance. As shown in Figure 3, training with the longest CoT consistently leads to higher preference accuracy across all benchmarks. **This demonstrates the effectiveness of length-based CoT filtering in improving reasoning quality.** It also highlights a trade-off between accuracy and inference efficiency, depending on which selection strategy is chosen.
>
> > W2. While the paper does present a new way of training a reward model, it is not original in the sense that it remixes well-known existing training techniques.
>
>
> We would like to clarify that while reasoning distillation (SFT stage of Think-RM) and rule-based RL are individually well known techniques, **their combination and algorithmic choices in Think-RM are specifically designed for reward modeling tasks**.
>
> For example, the SFT stage is essential because applying rule-based RL directly to an instruction tuned model fails to enable long-horizon reasoning. Since the final answer in reward modeling is binary, the model can often guess correct answer without generating meaningful reasoning steps. This differs from the existing practice for training reasoning LLMs (e.g., math reasoning), where rule-based RL can be applied directly to a base model without the SFT stage. In that case, because the final answer is not binary, the model must generate sufficient reasoning to reach the correct solution.
>
> We further apply rule-based RL after the SFT stage to **prune redundant reasoning traces, which is fundamentally different from the purpose of rule-based RL in DeepSeek-R1 that instead encourages progressively longer responses**.
>
> While long CoT reasoning has been explored for math and coding, we are the first to introduce long-horizon reasoning for reward modeling tasks and show that it is effective in this domain as well. **Additionally, we propose a pairwise RLHF method that enables seamless integration of pairwise GenRM into the RLHF pipeline, which is a completely new approach**.
>
>
> > W3. There is no discussion of limitations.
>
> Thank you for pointing this out. One major limitation of Think-RM is that, similar to other vertically scaled methods, Think-RM's horizontal inference-time scaling introduces additional computational and time costs compared to models of the same size. However, this horizontally extended CoT delivers significantly better performance, surpassing current vertical scaling approaches. Moreover, we partially mitigate these costs by applying RL training to Think-RM (SFT), which removes verbose or redundant reasoning steps and reduces response length while retaining only the essential reasoning steps. Without inference-time scaling, achieving comparable performance would require substantially increasing model size, an approach that is less efficient and still struggles on reasoning-heavy tasks like RM-Bench, as shown below:
>
> | Model                            | RewardBench | RM-Bench   |
> |----------------------------------|-------------|------------|
> | Llama3.1-8B-Instruct             |  71.0       | 63.8       |
> | Llama3.1-70B-Instruct            |  84.0       | 65.5       |
> | Think-RM-Llama3.1-8B-Instruct    |  **86.3**   | **75.1**   |
>
> > W4. I find many of the algorithmic decisions, like using CoT length and the skew matrix, unmotivated.
>
> > Q2. I wasn't clear on why we need the skew-matrix? The authors say, "This formulation ensures that longer reasoning chains...result in smaller reward differences". I am unsure why we want this? Some clarification would be helpful.
>
> Regarding the motivation for using CoT length as a scaling factor for preference strength in the binary case, as noted in Lines 24–25 of the manuscript, when two candidate responses are easy to distinguish (i.e., large preference strength), excessive reasoning is unnecessary, and the CoT length is typically shorter. Conversely, when the responses are harder to distinguish (i.e., smaller preference strength), longer CoT reasoning is often required. Therefore, scaling by (1/CoT length) provides a heuristic way to calibrate different preference strength for binary outputs. As described in Line 201, this scaling is not needed when using multiclass Think-RM, which inherently addresses this issue.
>
> Regarding the skew matrix, we emphasize that this is not an algorithmic decision. For notational simplicity, we refer to it as a skew-symmetric matrix, which is a mathematical property of the pairwise preference strength matrix $D$. Specifically,
>
> $$
> D\in\mathbb{R}^{n\times n} \text{ is skew symmetric} \Leftrightarrow D=-D^T.
> $$
>
> As described in Lines 199 - 203, each element $d_{ij}$ represents $\text{preference on j-th response} - \text{preference on i-th response}$, while $d_{ji}$ represents $\text{preference on i-th response} - \text{preference on j-th response}$. By definition,
> $$
> d_{ij}=-d_{ji}
> $$
>
> for all $i,j$ in the same group. Therefore, $D$ is inherently skew-symmetric, and we use the term "skew matrix" purely for notational simplicity, not as a design choice requiring further motivation. We will clarify this in the updated version.
>
> > Q1. Do we always get long reasoning chains even if they are not required with Think-RM? Are there potentially other heuristics we could use instead of simply the longest chains to generate the training data?
>
> Yes. Similar to other reasoning models such as DeepSeek R1, Think RM typically produces long reasoning chains even when shorter reasoning would suffice. This behavior aligns with the goal of this work, which is to demonstrate that horizontal scaling of current GenRMs improves preference output quality on reasoning-heavy tasks without additional vertical scaling and to show how such models can be trained. **Incorporating dynamic reasoning capabilities into Think RM is a separate interest, and we will note this in our future work section**.
>
> Of course, **Think RM could also be extended with adaptive thinking budgets**, allowing it to adjust reasoning length based on task difficulty. State-of-the-art reasoning models like Gemini 2.5 and Claude 3.7 already use such budgets, learned during training, to limit the number of reasoning tokens used per task [1, 2].
>
> *References*
>
> *[1] Comanici, Gheorghe, et al. "Gemini 2.5: Pushing the frontier with advanced reasoning, multimodality, long context, and next generation agentic capabilities." arXiv preprint arXiv:2507.06261 (2025).*
>
> *[2] Anthropic. "Claude 3.7 Sonnet and Claude Code." Anthropic Blog (2025).*

---

> > ### Comment · Reviewer_KhzV · 2025-08-04
> >
> > I thank the authors for their response. I have a follow-up question: I am still curious if the authors believe there could be other good heuristics one could use within the think-rm framework.
> >
> > Does it _have_ to be the longest chain? If I understand correctly, the idea is that long chains are likely good (but verbose), and the RM can be finetuned with the rule-based rewards to remove this verbosity.
> >
> > But is it possible that there could be a heuristic that could remove the need for the rule-based step?

---

> > > ### Author Response · Authors · 2025-08-04
> > >
> > > Thanks for the follow-up question. We would like to clarify that the rule-based RL step in Think-RM is **not solely aimed at reducing verbosity**. It plays a **critical role in improving in-distribution (ID) task performance** while maintaining similar performance on out-of-distribution (OOD) tasks.
> > >
> > > For example, in Table 1 (binary case) of our manuscript, **Think-RM (SFT + RL)** outperforms **Think-RM (SFT)** by **7%** on ID task (HelpSteer2 dataset). Similarly, in the additional results provided in our response to Reviewer BTue, using a different backbone model (Qwen2.5-3B-Instruct), the RL step yields a **4% improvement** on ID task (HelpSteer2 dataset). This demonstrates that RL training is **essential to surpass other baselines in both ID and OOD settings, with the additional benefit of reducing verbosity in reasoning traces**.
> > >
> > > The primary goal of our work is to show that horizontal scaling of current GenRMs improves preference output quality on reasoning-heavy tasks while remaining competitive on in-distribution tasks without requiring additional vertical scaling, and to demonstrate how such models can be trained effectively.
> > >
> > > Regarding alternative heuristics, we have not found other effective methods for filtering long CoT data. Our experiments (Section 4.3) indicate that **length-based filtering is informative**, as it allows a trade-off between accuracy and inference efficiency. Exploring other filtering strategies would be an interesting direction for future research but is beyond the current scope of this work.

---

> > > ### Author Response · Authors · 2025-08-06
> > >
> > > Dear Reviewer KhzV,
> > >
> > > We would like to kindly follow up to see whether our response has addressed your question regarding the possibility of removing subsequent RL training by exploring different filtering methods. Please let us know if you have any further questions or if there are points that remain unclear.
> > >
> > > Thank you,
> > >
> > > The Authors

---

> > > > ### Comment · Reviewer_KhzV · 2025-08-06
> > > >
> > > > You have addressed my concerns. I am willing to increase my score.

---

> > > > > ### Author Response · Authors · 2025-08-06
> > > > >
> > > > > We sincerely appreciate your active engagement in the discussion and are delighted to learn that our response has addressed your concerns. We will clarify the raised points in the updated version of the manuscript and are deeply grateful for the time and effort you invested in reviewing our paper.

---

### Official Review · Reviewer_XU3w · 2025-07-01

**Clarity:** 3
**Significance:** 2
**Originality:** 2
**Rating:** 4
**Confidence:** 4

**Summary:**

1. This paper introduces a two-stage training framework that enables long-horizon reasoning in generative reward models by modeling an internal thinking process.
2. It also introduces a pairwise RLHF pipeline that trains policies directly on pairwise preference signals using its reward model.

**Questions:**

1. I would like to understand the advantages and disadvantages of this framework compared to other long-reasoning-based reward models, such as DeepSeek-GRM [1], RM-R1 [2], and R3 [3]. Given the close release dates of these works, my question is more for context and curiosity—it will not heavily influence my overall evaluation.

2. Why is long internal thinking necessary for reward modeling? In math reasoning tasks, the large space of candidate answers makes long reasoning helpful. However, in reward modeling, there are typically only 2 to 5 candidate responses. Is such long reasoning truly needed for these relatively simple tasks?

3. In addition, when generating long chain-of-thought (CoT) reasoning traces, the filtering criterion is to retain only those samples with correct preference predictions. Given the simplicity of the task and the small number of candidate answers, how can we ensure the quality and meaningfulness of the generated long CoT reasoning traces?

4. The vertical-scaling baseline takes a majority vote over 16 CoTs, while Think-RM fine-tunes on the single longest trajectory. What is the performance of CoT-GenRM vertical scaling when keeping only the answer with the longest CoT path?

[1] Inference-Time Scaling for Generalist Reward Modeling

[2] RM-R1: Reward Modeling as Reasoning

[3] R3: Robust Rubric-Agnostic Reward Models

**Ethical Concerns:**

["NO or VERY MINOR ethics concerns only"]

**Final Justification:**

The authors addressed most of my concerns in the rebuttal. I have increased my score to 4.

**Limitations:**

Efficiency. This method relies on much longer sequences, which leads to more training and inference time. In addition, rule-based reinforcement learning also adds instability.

**Quality:**

2

**Strengths And Weaknesses:**

Strengths:
1. The paper provides extensive empirical evaluations across multiple evaluation benchmarks.
2. The paper is well-written and organized, making the technical content easy to follow.

Weaknesses:
1. Efficiency. This method relies on much longer sequences, which leads to more training and inference time. In addition, rule-based reinforcement learning also adds instability.
2. Limited model scale. All experiments use only Llama-3.1-8B-Instruct, so it is unclear whether the method still helps on other backbones.
3. Limited baselines. The paper mainly compares with the reproduced baselines. Please also include strong public reward models such as  GPT-4o, QwQ-32B(which is used to generate CoT rationales in this paper), DeepSeek-GRM and powerful BT reward models like Skywork Reward model.

---

> ### Author Rebuttal · Authors · 2025-07-31
>
> > W1. Efficiency. ... In addition, rule-based reinforcement learning also adds instability.
> > L. Efficiency. ... In addition, rule-based reinforcement learning also adds instability.
>
> We appreciate the reviewer's concerns regarding the efficiency and instability of rule-based RL.
>
> ### 1. Regarding the concern about efficiency.
>
> Due to the limited rebuttal space, please refer to our response on Reviewer Lj4C's W.
>
>
> ### 2. Regarding the concern about instability of rule-based RL.
>
> As described in Lines 178–182 and 276–278, **applying rule-based RL to the SFT-trained model in fact stabilizes its behavior**. The SFT model, trained on rich but often verbose reasoning traces, learns via rule-based RL to preserve only the essential reasoning steps and discard unnecessary verbosity. This effect is reflected in Tables 1–3, where Think-RM (SFT + RL) outperforms Think-RM (SFT) while generating significantly less reasoning tokens.
>
> > W2. Limited model scale.
>
> We acknowledge the reviewer's concern regarding the limited backbone model. To address this, we have expanded our experiments by implementing Think-RM and other baselines using **Qwen2.5-3B-Instruct** as an additional backbone model. Now, Think-RM has been implemented based on both the Llama and Qwen family models, ranging from 3B to 8B. Due to limited computational resources, we are not able to provide much larger models at the current moment.
>
> The key results across various benchmarks, including HelpSteer2 (in-distribution tasks), HelpSteer3 (moderately shifted tasks), RewardBench, and RM-Bench (out-of-distribution tasks), are presented in our response to Reviewer BTue. As shown, our new results with Qwen are consistent with the results with Llama presented in Tables 1–3 of the manuscript.
>
>
> > W3. Limited baselines.
>
> To gain a deep understanding of different reward model training schemes, such as BT, CoT-GenRM, DeepSeek-GRM, and Think-RM, it is essential to compare them on a **unified dataset** and with **the same backbone model**. **Our objective is not to just list the highest-performing models by their accuracies, but to conduct a controlled and fair comparison that reveals in which cases each method is effective**.
>
> To broaden the baselines, we re-trained DeepSeek-GRM under exactly the same conditions as Think-RM and CoT-GenRM (same dataset, same long CoT generator (QwQ-32B), same backbone model (Qwen2.5-3B-Instruct)). Results for DeepSeek-GRM with vertical inference-time scaling $m=16$ are included in our response to Reviewer BTue. We also report QwQ-32B's performance in the same tables.
>
> The Skywork reward model is simply a BT reward model trained with a carefully chosen data mixture and postprocessing strategy, resulting in a training set 13x larger than ours. Because of this discrepancy, a direct comparison with Think-RM would be less informative (just for reference, we provide additional RM-Bench results in the table below). Instead, in Tables 1–4 of the manuscript, we report a BT counterpart trained on the same dataset and backbone model as Think-RM for a fair comparison.
>
> | Model                            | RM-Bench  |
> |----------------------------------|------|
> |Skywork-Reward-Llama-3.1-Instruct-8B|70.1 |
> | Think-RM-Qwen2.5-Instruct-3B          |  70.4   |
> | Think-RM-Llama3.1-Instruct-8B          |  **75.2**   |
>
>
> > Q1. I would like to understand the advantages and disadvantages of this framework compared to other long-reasoning-based reward models, such as DeepSeek-GRM [1], RM-R1 [2], ...
>
>
> 1. **DeepSeek-GRM** can be viewed as a variant of CoT-GenRM that focuses on **vertical inference-time scaling** rather than horizontal long reasoning-based reward models. It improves CoT-GenRMs by prompting the model to generate instance-specific evaluation criteria adaptively, along with the critic and final answer. This leads to **generating diverse CoT paths to the final answer, which is essential for vertical scaling**. As reported in their Table 2, DeepSeek-GRM performs inference with 32 parallel generations. Additionally, as shown in their Figure 7, its response length is typically around 200–300 tokens, significantly shorter than Think-RM, which generates over 1000 tokens for challenging tasks. We note that **this adaptive evaluation criteria generation approach could also be incorporated into Think-RM**, as it currently uses prefixed evaluation criteria as described in Lines 144–146 of the manuscript.
>
> 2. **RM-R1**, similar to DeepSeek-GRM and different from Think-RM, uses adaptive evaluation rubric generation. However, like Think-RM, RM-R1 treats reward modeling as a reasoning problem. As shown in their Table 11, its reasoning relies on an explicit answer chain with a fixed output format (e.g., the model is expected to fill the format ```<type>...</type><rubric>...<justify>...</justify></rubric><eval>...</eval><answer>...</answer>```), which can restrict the model's natural reasoning process. In contrast, Think-RM imposes no such restrictions but only uses ```<think>...</think>``` so as not to prevent its inherent reasoning process. This freedom is essential for enabling advanced reasoning capabilities (see Figure 1 in our manuscript). As a result, according to their Figure 8, RM-R1's reasoning trace is approximately 600–800 tokens, still shorter than Think-RM's long-horizon reasoning. This difference contributes to Think-RM's superior performance on both RewardBench and RM-Bench, despite RM-R1 being trained on 42K samples compared to only 6K for Think-RM. As shown below, Think-RM based on the Llama3.1-8B-Instruct model outperforms RM-R1 based on comparably sized Qwen models, achieving significantly better results.
>
> | Model                            | RewardBench | RM-Bench |
> |----------------------------------|------|------|
> |RM-R1-DeepSeek-Distilled-Qwen-7B|80.1 | 72.4
> | RM-R1-Qwen2.5-Instruct-7B           |   85.2    | 70.2|
> | Think-RM-Llama3.1-Instruct-8B          |  **86.35**   | **75.06**|
>
> Beyond this, only Think-RM considers the downstream use of GenRMs within RLHF. Existing methods have yet to explore leveraging GenRM preference signals directly, instead relying on numerical reward scores. Our work takes this a step further by proposing a **pairwise RLHF pipeline that integrates GenRMs seamlessly, using their preference verdicts directly for policy optimization**.
>
>
>
> > Q2. Why is long internal thinking necessary for reward modeling? ... Is such long reasoning truly needed for these relatively simple tasks?
>
> We are not entirely sure what the reviewer means by```in math reasoning tasks, the large space of candidate answers makes long reasoning helpful.``` However, we believe that in math reasoning tasks, the ground-truth step-by-step reasoning path that connects the problem to the solution is inherently long; thus, long reasoning is helpful as it can better match this structure. As described in Lines 52–56 of the manuscript, in reward modeling for domains such as math, coding, adversarial prompts, or multi-turn contexts, the inherent step-by-step judgment path that connects the context and candidate responses to the final verdict is also long, making long-horizon reasoning in GenRM essential.
>
> This is supported by our main experiment results. As noted in Lines 291–293 of the manuscript, Think-RMs outperform CoT-GenRMs by over 10% and 5% in the **Chat Hard** and **Reasoning** subcategories of RewardBench, and by 12% in the **Math** domain of RM-Bench.
>
> If the reviewer would like to see real examples demonstrating the necessity of using long reasoning for reward modeling, we refer them to samples in RM-Bench [1], particularly those in the "math" or "code" domains, available via the dataset link below.
>
> *References*
>
> *[1] Liu, Yantao, et al. "RM-Bench: Benchmarking Reward Models of Language Models with Subtlety and Style." The Thirteenth International Conference on Learning Representations. https://huggingface.co/datasets/THU-KEG/RM-Bench*
>
> > Q3. In addition, when generating long chain-of-thought (CoT) reasoning traces, the filtering criterion is to retain only those samples with correct preference predictions. ... how can we ensure the quality and meaningfulness of the generated long CoT reasoning traces?
>
> This is an important point. Since the candidate answers are limited to A>B, A<B, or tie, it is possible that the reasoning trace may not always be fully accurate and might simply produce the correct preference label. As described in Section 4.3 (Ablation Study), we study a length-based reasoning trace selection strategy and show that selecting longer CoTs outperforms selecting shorter CoTs in terms of performance, with the trade-off of increased latency. We agree this is an interesting direction and will leave a more thorough investigation for future work.
>
> > Q4. The vertical-scaling baseline takes a majority vote over 16 CoTs, while Think-RM fine-tunes on the single longest trajectory. What is the performance of CoT-GenRM vertical scaling when keeping only the answer with the longest CoT path?
>
> The table below compares CoT-GenRM with vertical scaling, which uses the preference label from the longest CoT path, against majority-vote-based CoT-GenRM. As shown, the longest-path approach performs significantly worse than majority voting, indicating that for explicit CoT paths, the longest reasoning trace is not always of high quality.
>
>
> | Model          | HelpSteer2 | HelpSteer3 | RewardBench | RM-Bench |
> |----------------|------|------|------|--------|
> |  CoT-GenRM (**Longest Length**)
> w/ vertical inference-time scaling (m = 16)                                             | 70.74     |  67.97    |  72.75   | 63.90  |
> | CoT-GenRM (**Majority Voting**)
> w/ vertical inference-time scaling (m = 16) |  **74.43**   |   **70.34**  |  **75.18**    |  **65.63**      |

---

> > ### Author Response · Authors · 2025-08-06
> >
> > Dear Reviewer XU3w,
> >
> > We would like to kindly follow up regarding our rebuttal. We are available to answer any further questions or clarify any points that may remain unclear. We believe an active discussion with the reviewer would be invaluable for a thorough evaluation of the paper.
> >
> > Thank you,
> >
> >
> > The Authors

---

> > ### Comment · Reviewer_XU3w · 2025-08-06
> >
> > Thanks for your response. I still have a few follow-up questions:
> >
> > [A1] Regarding the concern about efficiency.
> > The baselines presented in the table are not convincing, as they are simply base models without any reward model training. Moreover, the comparison between Think-RM and CoT-GenRM with vertical scaling (which produces 3x to 4x more tokens) seems unfair. Across all reported settings, the performance gains from vertical scaling with m=16 are minimal (less than 1%), suggesting that either the additional tokens are not effectively contributing to performance.
> >
> > [A3-4] Additional experiments about model scale and baselines.
> > Thanks for the additional results. One minor question is that the average lengths for both CoT-GenRM(m=16) and Thank you for providing additional results. One clarification: under the Qwen2.5-3B-Instruct setting, the average output lengths of CoT-GenRM(m=16) and DeepSeek-GRM(m=16) are less than half of that of Think-RM. This trend is the opposite of what is reported in the main paper. Could you elaborate on why this discrepancy occurs?
> >
> > [A6] Why is long internal thinking necessary for reward modeling?
> > Thanks for your response. From my perspective, the reward model’s main responsibility is to evaluate the correctness or quality of a given answer, rather than to re-derive the solution through full reasoning. I am open to this question.
> >
> > To my understanding, the key difference between Think-RM (SFT) and CoT-GenRM (model-generated, without vertical scaling) lies in the length of the training samples—Think-RM is consistently trained on the longest CoT traces, while CoT-GenRM does not have any preference. Given this, it is confusing to observe a substantial performance gap—up to 7% in accuracy, as shown in Table 2, Table 3, and Figure 3. Can you explain it more?

---

> > > ### Author Response · Authors · 2025-08-08
> > >
> > > Dear Reviewer XU3w,
> > >
> > > As we are approaching the end of the discussion period, we would like to kindly follow up to see whether our response has addressed your follow-up questions. Please let us know if you have any further questions or if there are points that remain unclear.
> > >
> > > Thank you,
> > >
> > > The Authors

---

> > > ### Author Response · Authors · 2025-08-08
> > >
> > > Dear Reviewer XU3w,
> > >
> > > With only a few hours left in the reviewer–author discussion period, our discussion has not yet been finalized, and we are still awaiting your response. If further clarification is needed, please let us know, or kindly let us know whether your questions have been addressed.
> > >
> > > Thank you,
> > >
> > > The Authors

---

> ### Author Response · Authors · 2025-08-06
>
> We appreciate the reviewer's thoughtful follow-up questions.
>
> > [A1-1] Regarding the concern about efficiency. The baselines presented in the table are not convincing, as they are simply base models without any reward model training.
>
> We would like to clarify that CoT-GenRM (model-generated) and CoT-GenRM (ground truth) are **not base models without training** as described in Lines 231-240 of the manuscript. Both are trained using Equation (8) in [1]. For example,
>
> * CoT-GenRM (model-generated) uses explicit CoT rationales generated by the model.
>
> * CoT-GenRM (ground truth) uses human-written rationales, which were further refined by expert researchers, making it a **very strong baseline**.
>
> Additionally, following your suggestion, we included **another trained baseline**, DeepSeek-GRM, for further comparison in the additional experiment results.
>
>
>
> > [A1-2] Moreover, the comparison between Think-RM and CoT-GenRM with vertical scaling (which produces 3x to 4x more tokens) seems unfair. Across all reported settings, the performance gains from vertical scaling with m=16 are minimal (less than 1%), suggesting that either the additional tokens are not effectively contributing to performance.
>
>
> We understand that using m=16 may appear unfair. However, we chose m=16 to align with prior works on GenRMs [1, 2, 3], which use m=32. **To avoid exaggerating the number of generated tokens while still maintaining comparability with previous literature**, we conservatively selected m=16 for our experiments (rather than m=32).
>
> Alternatively, we could search for the optimal m, defined as the point where performance begins to plateau (with m on the x-axis and performance on the y-axis), and present that result. However, **this setting would be unfair to Think-RM**, as vertical scaling would then benefit from an additional inference-time parameter that requires tuning, whereas Think-RM does not rely on such optimization.
>
> Furthermore, the marginal gains from vertical scaling with m=16 highlight a limitation of CoT-GenRM. For vertical scaling, encouraging diverse judgments is crucial, and methods like DeepSeek-GRM aim to improve this diversity.
>
> > [A3-4] Additional experiments about model scale and baselines. Thanks for the additional results. One minor question is that the average lengths for both CoT-GenRM(m=16) and DeepSeek-GRM(m=16) are less than half of that of Think-RM. This trend is the opposite of what is reported in the main paper. Could you elaborate on why this discrepancy occurs?
>
> Thank you for pointing this out. In Tables 1-3, we report the total number of generated tokens across parallel generation, which is correct. On the other hand, in the additional experiment results, we report the average number of generated tokens across parallel generation, i.e., the average length per path, rather than the total across all m paths. To obtain the correct total token count, the reported numbers in the additional experiment results should be multiplied by m. We will update these numbers accordingly in the revised version.
>
>
> **Reference**
>
>
> [1] Mahan, Dakota, et al. "Generative reward models." arXiv preprint arXiv:2410.12832 (2024).
>
> [2] Zhang, Lunjun, et al. "Generative verifiers: Reward modeling as next-token prediction." arXiv preprint arXiv:2408.15240 (2024).
>
> [3] Yu, Yue, et al. "Self-generated critiques boost reward modeling for language models." arXiv preprint arXiv:2411.16646 (2024).

---

> ### Author Response · Authors · 2025-08-06
>
> > [A6] Why is long internal thinking necessary for reward modeling? Thanks for your response. From my perspective, the reward model’s main responsibility is to evaluate the correctness or quality of a given answer, rather than to re-derive the solution through full reasoning. I am open to this question.
>
> We agree that the primary role of a reward model is to evaluate the quality of a given response. Rather than this scalar reward model, GenRMs often take two candidate responses and output which is preferred. **For domains like math or code, this evaluation often requires deep understanding and inference, both of which demand significant reasoning**.
>
> Consider a case where the reward model compares two correct math solutions: one efficient, the other inefficient. To judge quality, the model must understand both solutions in depth (requiring reasoning) and infer the criteria by which they differ (e.g., efficiency, clarity, correctness). Furthermore, the model must reason about how to weigh these criteria to make a final judgment.
>
> > [Q1] To my understanding, the key difference between Think-RM (SFT) and CoT-GenRM (model-generated, without vertical scaling) lies in the length of the training samples—Think-RM is consistently trained on the longest CoT traces, while CoT-GenRM does not have any preference. Given this, it is confusing to observe a substantial performance gap—up to 7% in accuracy, as shown in Table 2, Table 3, and Figure 3. Can you explain it more?
>
> While it is true that training sample lengths typically differ between Think-RM and CoT-GenRM, this is **not** the main distinction. The critical difference lies in **the type of reasoning data** used for training. Think-RM is trained on internal thinking traces from the model. CoT-GenRM, on the other hand, is trained on explicit step-by-step rationales, which are typically much shorter and cleaner.
>
> For our experiments, we train Think-RM using QwQ-32B's internal thinking trace before it outputs the final step-by-step rationale and preference verdict. In contrast, we train CoT-GenRM using QwQ-32B's final step-by-step rationales after this thinking.
>
> Internal thinking traces are generally longer because they include exploration, self-reflection, and discarded reasoning paths (not just the final reasoning steps). This allows for **richer, more flexible reasoning styles**, as shown in Figure 1 of our manuscript, but also **introduces significant noise and verbosity**. As a result, Think-RM trained only with SFT tends to produce noisy and verbose reasoning traces, and underperforms on easier (in-distribution) tasks (see Table 1) compared to the baselines.
>
>
> We **address this issue through additional RL training** on the same prompt set used in SFT stage. Furthermore, we are the first propose a practical pairwise RLHF framework that seamlessly integrates GenRMs into entire RLHF training pipeline. This setup reflects the reward model's full role in guiding the training process of the policy, **which was missing** in previous and concurrent works.

---

### Official Review · Reviewer_BTue · 2025-07-03

**Clarity:** 3
**Significance:** 2
**Originality:** 3
**Rating:** 5
**Confidence:** 3

**Summary:**

The paper addresses a key issue in RLHF: creating reliable reward models, especially when data is scarce or tasks need deep reasoning. Conventional BT reward models are easy to train but often over-fit and are susceptible to reward hacking. Existing generative reward models (GenRMs) have short reasoning and produce pairwise preferences that don't fit well with standard RLHF pipelines.​

The authors present Think-RM, a generative reward-modeling framework that gives language models long-horizon, self-directed "internal thinking". It generates flexible, multi-thousand-token reasoning traces with self-reflection, hypothetical branching, and divergent analysis, solving the shallow reasoning problem of previous GenRMs.​

Tests show that an 8-billion-parameter Think-RM, trained on just 6k preference examples:​
* Tops RM-Bench, beating BT RMs and vertically-scaled GenRMs by about 8 percentage points.​
* Performs well on HelpSteer3-Preference, RewardBench and other out-of-distribution tasks, especially in "Chat-Hard", math and coding which need extended reasoning.​
* When used in the pairwise RLHF loop, leads to higher end-policy win rates than traditional pointwise RLHF with BT RMs.​

An ablation test shows that using the longest correct reasoning for warm-up is important, as it improves accuracy but makes outputs longer, showing a trade-off between depth and efficiency. Overall, Think-RM expands reward modeling from "vertical" ensemble scaling to "horizontal" depth, offering a practical way to use rich internal reasoning for preference-based RLHF.

**Questions:**

Can the authors evaluate Think‑RM against vertical GenRM settings and a human preference study on at least 100 prompts?

**Ethical Concerns:**

["NO or VERY MINOR ethics concerns only"]

**Quality:**

3

**Strengths And Weaknesses:**

​
This paper offers a compelling depth‑oriented alternative to today’s reward‑modeling practice: by pre‑training on very long chains‑of‑thought and then trimming them with rule‑based RL, an 8 B‑parameter model outperforms both classic BT and “vertical” GenRM baselines on every difficult benchmark the authors test—while feeding its pairwise logits directly into a new RLHF loop.  ​
​

The manuscript is easy to read. To my knowledge this is the first reward‑model paper that (i) mines and trains on multi‑kilotoken reasoning, (ii) prunes that reasoning with RL rather than heuristic truncation, and (iii) dispenses with pointwise score conversion in RLHF.  Earlier GenRMs limit each rationale to a few hundred tokens and rely on majority voting for robustness; Think‑RM flips that strategy.  The combination is novel even if each ingredient—long‑CoT mining, GRPO, pairwise PPO—exists in separate literature streams. Reward‑model brittleness is a bottleneck for RLHF deployment, especially when tasks demand deep reasoning and preference data are scarce.  By showing that horizontal scaling (thousands of tokens in one trace) beats the vertical self‑consistency trick that dominates today’s GenRMs, Think‑RM shifts the conversation away from ever‑larger majority‑vote ensembles.  ​
​

My only concern is that I'm afraid that human judge is needed for such RM paper. While the paper shows impressive results on various benchmarks, real-world applications of RLHF often involve subjective human preferences. Incorporating human judges in the evaluation process could provide a more comprehensive assessment of Think-RM's performance. For example, in scenarios such as content generation for creative writing or personalized advice, human judgment of the quality and appropriateness of the generated content is indispensable.

---

> ### Author Rebuttal · Authors · 2025-07-31
>
> > W1. My only concern is that I'm afraid that human judge is needed for such RM paper. While the paper shows impressive results on various benchmarks, real-world applications of RLHF often involve subjective human preferences. Incorporating human judges in the evaluation process could provide a more comprehensive assessment of Think-RM's performance. For example, in scenarios such as content generation for creative writing or personalized advice, human judgment of the quality and appropriateness of the generated content is indispensable.
>
> > Q1. Can the authors evaluate Think‑RM against vertical GenRM settings and a human preference study on at least 100 prompts?
>
> We appreciate the reviewer's valuable comment. Since conducting a human preference study is costly, we use Gemini 2.5- Pro as a practical proxy for human judges to evaluate the quality of preference judgments. Specifically, we evaluate the depth and breadth of rationales generated by Think‑RM compared to CoT‑GenRM on a reward modeling task using prompts from MT‑Bench‑Hard, a challenging subset of MT‑Bench designed to test instruction-following and reasoning abilities in multi-turn interactions. Under this evaluation, Think‑RM achieves a 73.3% win rate, demonstrating strong alignment with human-preferred preference reasoning. Below is a sample judgment from Gemini 2.5 Pro (Judge A: Think-RM vs. Judge B: CoT-GenRM):
>
> ```
> Judge A provides a clear and accurate analysis of both assistant responses. Their reasoning correctly identifies that Assistant A fulfilled the user's request by writing a complete short story, whereas Assistant B only provided a single additional sentence, failing to deliver a story. Judge A methodically breaks down why Assistant A's response is superior across several criteria, including helpfulness, coherence, and complexity.
>
> In contrast, Judge B's evaluation is fundamentally flawed. It appears to have completely confused the two assistant responses, attributing the qualities of Assistant A's story (e.g., "full, engaging narrative arc," "first-person narrative," "atmospheric description") to Assistant B, and describing Assistant A's response as being only one sentence long. Because its entire reasoning is based on this factual error, its conclusion is incorrect and its judgment is unreliable.
>
> Judge A's evaluation is superior due to its accuracy, logical consistency, and correct interpretation of the provided inputs.
> ```
>
> > Beyond what was requested in the review, all experiments in the manuscript were soley conducted using Llama‑3.1‑Instruct‑8B. We have now additionally obtained results with Qwen‑2.5‑Instruct‑3B, which are included here. These new results are consistent with the Llama‑based results reported in Tables 1–3 of the manuscript.
>
> ### HelpSteer2 (Qwen2.5-3B-Instruct)
> | Model                                              | Validation | Avg. Len |
> |----------------------------------------------------|------------|----------|
> | Base                                               |    53.84   | 262.74   |
> | BT RM                                              |      71.01      |      -    |
> | CoT-GenRM
> w/ vertical inference-time scaling (m = 16) |   74.43     |    377.09      |
> | DeepSeek-GRM
> w/ vertical inference-time scaling (m = 16)|  72.74          | 160.49         |
> | Think-RM (SFT)                                     |   71.31   | 1061.05      |
> | Think-RM (SFT + RL)                                |    **75.99**   |   836.62       |
> | QwQ-32B                                               |   78.69    | 760.82        |
>
>
> ### HelpSteer3 (Qwen2.5-3B-Instruct)
> | Model                                              | Code | General | Multilingual | Stem | AVG | Avg. Len |
> |----------------------------------------------------|------|---------|--------------|------|-----|----------|
> | Base                                               | 62.27| 53.61   |     54.24    |     53.70  |  55.68   |    365.49      |
> | BT RM                                              |   70.14   |   63.39      |      64.20        | 69.09     |   65.99  |    -      |
> | CoT-GenRM
> w/ vertical inference-time scaling (m = 16) |    76.16   |      67.98     |    73.94          |     63.99  |    **70.34**   |   388.08       |
> | DeepSeek-GRM
> w/ vertical inference-time scaling (m = 16)|            73.15 | 65.74       | 72.73 | 66.67 | 68.72 |162.6 |
> | Think-RM (SFT)                                     | 73.15     |   65.74      |        74.70      |     67.49  |    69.17  | 1230.71  |
> | Think-RM (SFT + RL)                                |       75.93        |     67.16 |    71.06  | 67.70  | 69.87 | 849.66     |
> | QwQ-32B                                             |  84.49    |     71.80 | 81.82    | 67.49    | 75.83    |  863.27      |
>
>
> ### RewardBench (Qwen2.5-3B-Instruct)
> | Model                                              | Chat | Chat Hard | Reasoning | Safety | AVG | Avg. Len |
> |----------------------------------------------------|------|-----------|-----------|--------|-----|----------|
> | Base                                               |   73.74    |    48.03        |    60.52       |    69.86     |   63.60    |    321.59      |
> | BT RM                                              |  87.43    |  62.50         |   73.74       |   75.95      |  74.90   |    -      |
> | CoT-GenRM
> w/ vertical inference-time scaling (m = 16) |  94.27    |     56.47       |      71.16       |    79.53    |  75.18   |   341.56       |
> | DeepSeek-GRM
> w/ vertical inference-time scaling (m = 16)|          93.3  | 54.71          | 69.41 | 77.09 | 73.74 |  160.18 |
> | Think-RM (SFT)                                     | 91.76     |      62.28      |     72.06       |      82.64   |  76.28    |    1844.35    |
> | Think-RM (SFT + RL)                                |     93.58    |    62.83        |      75.77     |   81.96   |   **78.56**   |   1172.42     |
> | QwQ-32B                                             |  96.93    |  82.24   | 97.62    | 89.59    | 93.45    | 1123.66       |
>
>
> ### RM-Bench (Qwen2.5-3B-Instruct)
> | Model                                              | Chat | Code | Math | Safety | AVG | Avg. Len |
> |----------------------------------------------------|------|------|------|--------|-----|----------|
> | Base                                               |  55.86    | 51.12      |  53.12    |   73.75   | 59.90  |    319.84      |
> | BT RM                                              |  54.61    |  54.19    |   58.52   |  70.09      |  61.24   |      -    |
> | CoT-GenRM
> w/ vertical inference-time scaling (m = 16) |  59.86    |   51.17   |  59.20    |  82.51      |  65.63   | 354.83       |
> | DeepSeek-GRM
> w/ vertical inference-time scaling (m = 16)|  61.46          | 50.88         | 55.95| 81.56|64.13 |160.51 |
> | Think-RM (SFT)                                     |  62.88    |  51.85   |    64.59|   87.30    |    69.78  |   1884.25      |
> | Think-RM (SFT + RL)                                |    60.42    |    52.73   |  66.54   |    87.05     | **70.39**    |  1169.78     |

---

### Official Review · Reviewer_Lj4C · 2025-07-03

**Clarity:** 4
**Significance:** 3
**Originality:** 2
**Rating:** 4
**Confidence:** 3

**Summary:**

This paper proposes a new training framework for the Generative Reward Model. In many benchmarks, Think-RM shows an improvement over other reward models. The paper also compares pairwise RLHF with pointwise RLHF, which shows a more effiective performance.

**Questions:**

Could the authors clarify the difference between the SFT phase of Think-RM and the COT-GenRM training, as their training objectives appear similar?

In Table 4, the performance gap between GenRM and Think-RM on HH-RLHF is minimal and requires further explanation.  And authors claim that the WR of Think-RMs is higher than COT-GenRMs, but LC is a more important metric in  AlpacaEval2.

Why is the number of independent judgments in Table 4 inconsistent with those in other tables?

**Ethical Concerns:**

["NO or VERY MINOR ethics concerns only"]

**Final Justification:**

The concern about computational and time costs, marginal performance improvements of Think-RM over CoT-GenRM in Table 1 and Table4 have been addressed.

**Limitations:**

The paper does not propose specific mechanisms to detect or correct hallucinated reasoning chains, which are common failure modes.

**Paper Formatting Concerns:**

N.A.

**Quality:**

3

**Strengths And Weaknesses:**

Strength:
The writing of this paper is clear and easy to follow.
The paper evaluates the proposed method accorss many benchmarks, including HelpSteer2-Preference, HelpSteer3-Preference, RewardBench, RM-Bench and AlpacaEval 2.

Weakness:
The model trained with Think-RM demostrates a longer response length in all tables, which may indicates more computational and time cost.
In Table 1 and Table 4, there are marginal improvements between Think-RM and COT-GenRM.

---

> ### Author Rebuttal · Authors · 2025-07-31
>
> > W. The model trained with Think-RM demostrates a longer response length in all tables, which may indicates more computational and time cost. In Table 1 and Table 4, there are marginal improvements between Think-RM and COT-GenRM.
>
>
> We appreciate the reviewer's observations regarding the computational cost and marginal performance improvements of Think-RM over CoT-GenRM in Tables 1 and 4.
>
> ### 1. Regarding the concern about computational and time costs.
>
> We would like to clarify that **the longest response lengths across all tables are produced by CoT-GenRM with vertical scaling**, which generates 3x to 4x more tokens than Think-RM during inference. Despite this, **Think-RM outperforms CoT-GenRM with vertical scaling on most benchmarks, achieving 10%, 5%, and 12% improvements on reasoning-heavy tasks**, such as the Chat Hard, Reasoning, and Math subcategories of RewardBench and RM-Bench.
>
> Whether along the vertical or horizontal axis, inference-time scaling for solving complex problems is an active area of research across various LLM applications, including reasoning LLMs, tool-augmented LLMs, self-critique LLMs, and multi-agent debate (MAD). In this line of research, **we are the first to show that horizontally scaling current CoT-GenRMs further improves the quality of preference outputs, even without generating multiple CoT paths, and to demonstrate how such models can be trained**.
>
> Although horizontal inference-time scaling introduces additional computational and time costs compared to models of the same size, **this horizontally extended CoT yields significantly better performance**. Moreover, we partially address this limitation by applying RL training to Think-RM (SFT) to eliminate verbose or redundant reasoning steps, reducing response length while preserving only the essential steps. Without inference-time scaling, **achieving similar performance would require scaling the model size substantially, which is far less efficient and still struggles with reasoning-heavy tasks like RM-Bench**, as shown below:
>
> | Model                            | RewardBench | RM-Bench   |
> |----------------------------------|-------------|------------|
> | Llama3.1-8B-Instruct             |  71.0       | 63.8       |
> | Llama3.1-70B-Instruct            |  84.0       | 65.5       |
> | Think-RM-Llama3.1-8B-Instruct    |  **86.3**   | **75.1**   |
>
> ### 2. Regarding the concern about marginal performance improvements of Think-RM over CoT-GenRM in Table 1.
>
> For in-distribution tasks (Table 1), since the problems exhibit recurring patterns similar to the model's training examples, pattern matching or shallow reasoning is often sufficient to produce correct preference outputs. Consequently, BT-RM and CoT-GenRMs have an inherent advantage on these tasks. We address this by introducing RL training (stage 2 in Think-RM), where **additional RL fine-tuning on the same prompt set from the SFT stage significantly improves performance on in-distribution tasks** while maintaining similar performance on out-of-distribution (OOD) tasks. **For example, in Table 1 (binary case), Think-RM (SFT + RL) improves by 7% over Think-RM (SFT) and by 4% in the table provided in our response to Reviewer BTue, which uses a different backbone model (Qwen2.5-3B-Instruct)**, performing competitively with or outperforming BT-RM and CoT-GenRMs even on in-distribution tasks. Therefore, we argue that the performance gains, though seemingly marginal in Table 1, are a strength of the proposed method.
>
> ### 3. Our response regarding Table 4 is in our response on reviewer's Q2 below.
>
>
> > Q1. Could the authors clarify the difference between the SFT phase of Think-RM and the COT-GenRM training, as their training objectives appear similar?
>
> While the training objectives are indeed the same, the SFT phase of Think-RM is anotherly described as **reasoning distillation** [1, 2]. The key distinction lies in the training data used.
>
> In current reasoning LLMs, given an input $x$, the model typically generates an internal thinking trace $r$, followed by a final explicit CoT response $y$. Think-RM is trained on the pair $(x, r)$, focusing on learning internal thinking process described in Figure 1 in the manuscript. In contrast, CoT-GenRM is trained on $(x, y)$, focusing on learning external CoT traces.
>
> *References*
>
> *[1] Guo, Daya, et al. "Deepseek-r1: Incentivizing reasoning capability in llms via reinforcement learning." arXiv preprint arXiv:2501.12948 (2025).*
>
> *[2] Kang, Minki, et al. "Distilling llm agent into small models with retrieval and code tools." arXiv preprint arXiv:2505.17612 (2025).*
>
> > Q2. In Table 4, the performance gap between GenRM and Think-RM on HH-RLHF is minimal and requires further explanation. And authors claim that the WR of Think-RMs is higher than COT-GenRMs, but LC is a more important metric in AlpacaEval2.
>
> As noted in Line 313 of the manuscript, CoT-GenRM and Think-RM achieve nearly identical preference accuracy on the HH-RLHF test set (64.42 vs. 65.03). This indicates that HH-RLHF is not sufficiently discriminative to reveal the full advantage of Think-RM over CoT-GenRM, which explains the marginal improvement reported in Table 4. Although LC WR is indeed emphasized in AlpacaEval2, WR is still a critical complementary metric. On this metric, Think-RM significantly outperforms CoT-GenRM despite the slight difference in HH-RLHF test set accuracy. Moreover, we would like to emphasize that our contribution is twofold: **1) Think-RM training methodology**, and **2) Pairwise RL approach** that enables seamless integration of GenRMs into RLHF pipelines. **The primary purpose of Table 4 is to demonstrate the effectiveness of our pairwise RLHF method compared to traditional pointwise RLHF with scalar rewards**. As shown in Table 4, our pairwise RLHF framework significantly outperforms traditional pointwise RLHF with BT models, bridging the gap between expressive GenRMs and practical RLHF pipelines.
>
> > Q3. Why is the number of independent judgments in Table 4 inconsistent with those in other tables?
>
> For the reward modeling benchmarks in Tables 1-3, we use 16 independent runs to align with prior work [1, 2] on vertical test-time scaling, which typically uses values greater than 8. However, in Table 4, we keep the total number of generated tokens during inference roughly comparable between CoT-GenRM and Think-RM due to the computational bottleneck. **Since CoT-GenRM with vertical scaling (m=16) generates 3–4 times more tokens than Think-RM in Tables 1-3, we reduce the number of runs to m=4**. We will make this clarification in the updated version of the manuscript.
>
> *References*
>
> *[1] Yu, Yue, et al. "Self-generated critiques boost reward modeling for language models." NAACL 2025 (2024).*
>
> *[2] Liu, Zijun, et al. "Inference-time scaling for generalist reward modeling." arXiv preprint arXiv:2504.02495 (2025).*

---

> > ### Author Response · Authors · 2025-08-06
> >
> > Dear Reviewer Lj4C,
> >
> > We would like to kindly follow up regarding our rebuttal. We are available to answer any further questions or clarify any points that may remain unclear. We believe an active discussion with the reviewer would be invaluable for a thorough evaluation of the paper.
> >
> > Thank you,
> >
> > The Authors

---

> > ### Comment · Reviewer_Lj4C · 2025-08-06
> >
> > Thanks for the reply. You have addressed my concerns. I would like to raise my score.

---

> > > ### Author Response · Authors · 2025-08-06
> > >
> > > We are sincerely pleased to hear that our explanation addressed your concerns. Your thoughtful and detailed feedback has been invaluable in refining our work, and we deeply appreciate the time and effort you invested in reviewing our paper.

---

### Comment · Area_Chair_6j4m · 2025-08-04
**Engage in Author-Reviewer Discussions**

Dear reviewers,

If you haven't done so already, please click the 'Mandatory Acknowledgement' button and actively participate in the rebuttal discussion with the authors after carefully reading all other reviews and the author responses.

Thanks,
AC

---

### Note · Authors · 2025-08-14

We express our deepest gratitude to the reviewers for their thorough feedback and for recognizing the clarity of our presentation (Reviewers Lj4C, BTue, XU3w) and our main contributions, which include:

1. **First paper** to flip GenRM's scaling axis from vertical to horizontal, and to introduce the novel pairwise RLHF method (Reviewer BTue).
2. **Extensive experiments** across many benchmarks, showing improvements over vertical scaling on both in-distribution (ID) and out-of-distribution (OOD) tasks, with especially large gains on reasoning-heavy and OOD tasks (Reviewers Lj4C, BTue).

Below, we summarize the major concerns and our responses:

**Efficiency (Reviewers Lj4C, XU3w)**

Think-RM intentionally uses additional tokens to boost performance on challenging preference judgment tasks, aligned with concurrent inference-time scaling work in reasoning and agentic LLMs. Scaling model size is an alternative, but we show it is less efficient and remains weak on reasoning-heavy tasks.

**Rule-based RL (Reviewers Lj4C, XU3w, KhzV)**
After SFT stage, Think-RM often shows redundant and verbose reasoning and underperforms on easy tasks (e.g., ID tasks). We address this with additional RL training, rather than relying on heuristic methods, which improves ID performance by 7% (Llama) and 4% (Qwen) while maintaining strong OOD results (10%, 5%, 12% on Chat Hard, Reasoning, Math). RL exploration naturally prunes redundancy, increasing stability.

**Limited Model Scale (Reviewer XU3w)**
We added Qwen2.5-3B-Instruct results for all benchmarks. Findings are consistent with Tables 1-3, showing Think-RM generalizes across model sizes and families.

**Limited Training-Based Baselines (Reviewer XU3w)**
As described in Lines 231-240, we corrected Reviewer XU3w’s misunderstandings by clarifying that all baselines are trained RMs and that the key difference between CoT-GenRM and Think-RM lies in the type of reasoning data used for training, not in data length. Additional experiments include another recent trained baseline, DeepSeek-GRM.

Reviewers Lj4C, KhzV, and BTue confirmed their concerns were resolved, with Lj4C and KhzV explicitly indicating they will raise their scores, and BTue maintaining a positive score. Although Reviewer XU3w did not respond further, we respectfully believe our rebuttals have addressed their concerns.

We believe these clarifications and results establish Think-RM as a significant contribution to advancing generative reward modeling.

---

### Decision · Program_Chairs · 2025-09-17

**Decision:**

Accept (poster)

**Comment:**

This paper proposes a novel generative reward model that generates long-horizon rationales for preference decision between two candidate outputs. In specific, it trains the proposed reward model, Think-RM, by SFT on long CoT data followed by rule-based RL. Then, it applies to a modified pairwise GRPO. Experimental results show that the proposed reward model for horizontal inference-time scaling outperforms previous vertical inference-time scaling with generative reward models on RM evaluation benchmarks, and the proposed pairwise RLHF performs better than pointwise RLHF.

Overall, the long-horizon reasoning instead of previous vertical scaling for preference reward estimation seem to be interesting and techically sound. A modified GRPO for pairwise RLHF with the proposed reward model also seems to be somewhat novel. Most concerns raised by the reviewers are well addressed by the authors, including a lack of experiments regarding different backbones and baselines, during the rebuttal. Based on the consensus of all positive ratings between the reviewers, I would recommend the paper to be accepted.

However, some concerns still need to be further resolved. First, even though the main focus on the results in Table 4 is to demonstrate the benfits in using pairwise RLHF, the synergetic improvement by both the pairwise RLHF and Think-RM would be necesseary with more diverse LLM benchmarks, from the persepective of the praticality. In addition, regarding the Think-RM training framework, more diverse ablations and analyses on the use of longest CoT filtering and rule-based RL would be necessary. For example, unlike in Figure 3, one could consider mixing long and short CoTs for SFT, or incorporating factors such as a length penalty into the reward function during RL.